# Tracing fossil-based plastics, chemicals and fertilizers production in China

Meng Jiang [1,2,9], Yuheng Cao [1,9], Changgong Liu[3], Dingjiang Chen[1,4], Wenji Zhou[5], Qian Wen[6], Hejiang Yu[1], Jian Jiang[1], Yucheng Ren[1], Shanying Hu [1,4], Edgar Hertwich [2] ✉ & Bing Zhu [4,7,8] ✉

Phasing down fossil fuels is crucial for climate mitigation. Even though 80−90% of fossil fuels are used to provide energy, their use as feedstock to produce plastics, fertilizers, and chemicals, is associated with substantial $CO_2$ emissions. However, our understanding of hard-to-abate chemical production remains limited. Here we developed a chemical process-based material flow model to investigate the non-energy use of fossil fuels and $CO_2$ emissions in China. Results show in 2017, the chemical industry used 0.18 Gt of coal, 88.8 Mt of crude oil, and 12.9 Mt of natural gas as feedstock, constituting 5%, 15%, and 7% of China's respective total use. Coal-fed production of methanol, ammonia, and PVCs contributes to 0.27 Gt $CO_2$ emissions ( ~ 3% of China's emissions). As China seeks to balance high $CO_2$ emissions of coal-fed production with import dependence on oil and gas, improving energy efficiency and coupling green hydrogen emerges as attractive alternatives for decarbonization.

Mitigating climate change demands fossil fuel phase-down, which is a complex task given their dual role not only as dominant energy sources—accounting for 80−90% of today's consumption[1], but also as essential raw materials. While the energy use of these fuels is often highlighted, the non-energy application, especially in the chemical industry, contributes substantially to carbon emissions[2]. Chemical production necessitates vast amounts of heat, steam, and electricity, leading to notable $CO_2$ emissions from the energy supply (energy-related emissions)[2]. Furthermore, process emissions, which arise directly from chemical reactions, are inherent and cannot be lessened without technological adjustments. This factor underscores why the chemical sector is labeled a "hard-to-abate" area. However, discussions often obscure or neglect the pivotal role of fossil fuels, as raw materials or feedstocks, in generating plastics, fertilizers, and other chemicals[3–5].

The discourse in energy and environmental studies frequently overlooks the distinct role of fossil hydrocarbons as raw materials. The term *fossil fuels* is used literally more often than *fossil resources* or *hydrocarbon resources*. Although extensive research has been conducted on the energy flows of fossil hydrocarbon utilization[6–11], the proportion used as feedstocks is unclear[2–5]. Even though fossil hydrocarbons are still predominantly used as energy sources, this practice will be phased down as the energy transition progresses. In contrast, the production of chemicals will drive the increase in demand for fossil hydrocarbons over the coming decades[1,12–15]. Modern energy systems extensively incorporate petrochemicals in components such as solar panels, wind turbine blades, batteries, building insulation, and electric vehicle parts[1]. Predictions indicate a potential tripling of global plastics production by 2060[16]. By 2050, petrochemical outputs could dominate global oil consumption[1]. In a transformative shift, many oil giants are pivoting from energy provision to material manufacturing[13,14,17], with the majority of new chemical complexes leveraging oil-to-chemical (OTC) technology to transform up to 70%–80% of crude oil into high-value chemicals[4,18]. Understanding the use of fossil hydrocarbons as feedstock will thus become increasingly

[1]Department of Chemical Engineering, Tsinghua University, Beijing, China. [2]Industrial Ecology Programme, Department of Energy and Process Engineering, Norwegian University of Science and Technology, Trondheim, Norway. [3]China Petroleum & Chemical Corporation (Sinopec), Beijing, China. [4]Institute for Circular Economy, Tsinghua University, Beijing, China. [5]School of Applied Economics, Renmin University of China, Beijing, China. [6]China National Petroleum & Chemical Planning Institute, Beijing, China. [7]The State Key Lab of Chemical Engineering, Department of Chemical Engineering, Tsinghua University, Beijing, China. [8]Energy, Climate, and Environment Program, International Institute for Applied Systems Analysis, Laxenburg, Austria. [9]These authors contributed equally: Meng Jiang, Yuheng Cao. ✉e-mail: edgar.hertwich@ntnu.no; bingzhu@tsinghua.edu.cn

crucial during the transition period as the proportion of chemical feedstocks in overall demand is rising globally[1].

In analyzing any material flow system, such as the chemical industry, the initial step involves identifying and quantifying mass flows to uncover the most important components[19]. However, a gap exists between macro-level modeling of material flows and carbon emissions related to chemical production. Environmental generalists and modelers are typically more concerned with the general environmental impacts and interactions between industries at the macroscale[1,4,20–22]. Chemical engineers, however, investigate the mass and energy balance of a typical technology and processes in a chemical plant. This divergence often results in incomplete assessments of fossil hydrocarbon feedstocks' material and energy dynamics. For example, while plastics are receiving an increasing amount of attention[23,24], the majority of existing research focuses on the use-phase[25–30] and general emissions[31–34], sidelining detailed production considerations (the non-energy employment of fossil fuels as feedstocks)[35,36]. Such insufficient understanding limits our capacity to see the overall landscape and the opportunities therein for carbon reduction, resource efficiency enhancement, and the fostering of circularity. Levi and Cullen[4] explored this field by mapping global chemical flows and suggesting expanding temporal, technological, and geographic resolution to identify trends and disparities in the global and regional chemical sector.

China's role in the global chemical sector is of critical importance due to the world's largest production volume[37,38], high fossil fuel consumption, and unique coal-chemical technology. While leading the world in both chemical and fossil fuel consumption, China's per capita chemical product consumption still lags behind high-income nations[39]. High-end product production, still reliant on imports, is not at par with demand in China's chemical domain[37]. Consequently, many institutions hold the view that China's chemical industry is still experiencing a period of rapid expansion[37,39]. Distinct from regions like Europe, North America, and the Middle East, which predominantly employ natural gas and crude oil, China extensively taps coal as both an energy source and chemical feedstock[40]. This practice amplifies the carbon intensity of China's chemical outputs[35]. Estimates show China's chemical industry accounted for 10–14% of China's total $CO_2$ emissions around 2017, equivalent to approximately 4% of global emissions[20,39,41,42]. However, limited studies have investigated the processes and flow from fossil feedstock to chemical products in China's chemical industry.

In this study, we present a tailored process-based material flow model for China's chemical industry, highlighting the interwoven mass and carbon flows within the expansive (gate-to-gate) production network. While global chemical flows have been mapped by Levi and Cullen[4], our research uniquely focuses on China's chemical industry, which stands out as the world's largest and is heavily reliant on coal. We further assess the process emissions arising directly from chemical production, alongside energy emissions stemming from the provision of heat, electricity, and steam, by using the coefficient-based carbon accounting method (scope 2 accounting). This allows us to pinpoint the production of primary chemical building blocks and identify the most carbon-intensive processes. Collecting foundational data for mass and carbon flow operations within the hydrocarbon-based chemical industry is challenging[4,43]. This study seeks to overcome these data limitations by sourcing and cross-verifying information from various resources, including the yearbook (China Chemical Industry Yearbook[44], Energy Statistics Yearbook[45]), disclosed industrial reports, other databases, and engagements with industry practitioners, all synthesized with an underpinning of chemical engineering knowledge. We also use Monte Carlo simulation[46] to explore potential deviations. All results presented in this study are based on data from the year 2017. Our analysis also underscores the challenges of decarbonizing China's chemical industry and anticipated shifts in the usage of hydrocarbon resources as primary materials. In essence, our analysis establishes a solid groundwork for future research to examine the broader implications of major structural changes on the energy, chemicals, plastics, and emissions nexus.

## Results

### Fuel Use and Feedstock Use

China's fossil fuel consumption patterns highlighted its reliance on coal and considerable import dependency on oil and gas. In 2017, China consumed 4.7 Gt (billion tons, equivalent to petagrams, Pg) of fossil hydrocarbons, including 3.9 Gt·yr$^{-1}$ of coal, 0.6 Gt·yr$^{-1}$ of crude oil, and 0.2 Gt·yr$^{-1}$ of natural gas, as shown in Fig. 1. China's natural abundance of coal contrasted with its insufficient oil and gas reserves, leading to the utilization landscape of coal and substantial import needs for crude oil and natural gas. Specifically, a substantial 93% of coal requirements were sourced domestically, aligning with China's rich coal deposits. Conversely, to satisfy its needs, China imported 70% of its crude oil and 37% of its natural gas consumption in 2017 (which varies slightly from 2018 real-time news data, showing 68%[47] oil and 39%[48] gas dependence, likely due to subsequent validation).

Our analysis distinguishes the roles of fossil hydrocarbons in energy versus feedstock for chemical production among the overall fossil fuel consumption in China. According to our material flow modeling, Fig. 1 illustrates the distribution of fossil hydrocarbon use in 2017. The majority of fossil hydrocarbons were used for energy, with coal at 95.4( ± 1)%, oil at 85.1( ± 0.2)%, and natural gas at 92.9( ± 1)% based on mass. In contrast, the chemical industry utilized 4.6( ± 1)% of the consumed coal, 14.9( ± 0.2)% of the consumed crude oil, and 7.1( ± 1)% of the consumed natural gas as feedstocks. Even though the current share of feedstocks in China (0.3 Gt·yr$^{-1}$ ( ± 10%)) was dwarfed by fuel use (4.3 Gt·yr$^{-1}$ ( ± 1%)), its emissions are considerable. Importantly, as the global energy sector moves away from fossil fuels, the demand for chemical feedstocks is projected to become an increasingly dominant force in hydrocarbon consumption[1,49].

### Mapping Chemical Production Flows

The chemical industry converts hydrocarbon feeds into primary chemicals, which serve as the building blocks for a myriad of products. The primary chemicals include ethylene, propylene, C4 olefins, BTX aromatics (benzene, toluene, and xylene), and methanol (from syngas for downstream chemical synthesis)[50]. In addition, ammonia synthesis, integral to fertilizer production, has profound implications for agricultural production and food security. As depicted in Fig. 2, our material flow analysis allows for the detailed tracking of these transformations, identifying the specific carbon (feedstock) sources for each intermediate and final chemical product. In 2017, China transformed 88.8 Mt·yr$^{-1}$ ( ± 1%) crude oil, 12.9 Mt·yr$^{-1}$ ( ± 16%) natural gas, and 0.18 Gt·yr$^{-1}$ ( ± 15%) coal as raw materials for chemical production.

Crude oil (petroleum) is the primary feedstock for the chemical industry. Various components of crude oil are broken down into smaller hydrocarbons through cracking and reforming processes (e.g., steam cracking, catalytic reforming, fluidized catalytic cracking, etc.). China converted 88.8 Mt·yr$^{-1}$ ( ± 1%) crude oil into 14.6 Mt·yr$^{-1}$ ( ± 3%) ethylene, 7.7 Mt·yr$^{-1}$ ( ± 29%) propylene, 4.2 Mt·yr$^{-1}$ ( ± 22%) C4 components, and 7.8 Mt·yr$^{-1}$ ( ± 22%) BTX aromatics. As Fig. 3 and Fig. 4 illustrate, the petroleum route is responsible for the vast majority of the production of these chemicals in China, contributing to 80( ± 2)% of ethylene production, 75( ± 2)% of propylene production, 100% C4 olefins production, and almost 100% BTX production.

China's chemical industry demonstrates a unique reliance on coal, distinguishing it from global norms due to its abundant coal reserves and limited supplies of natural gas and crude oil[51]. Our results show coal is the principal feedstock for producing methanol (89 ± 2%) and ammonia (76 ± 6%), two pivotal compounds in China's chemical

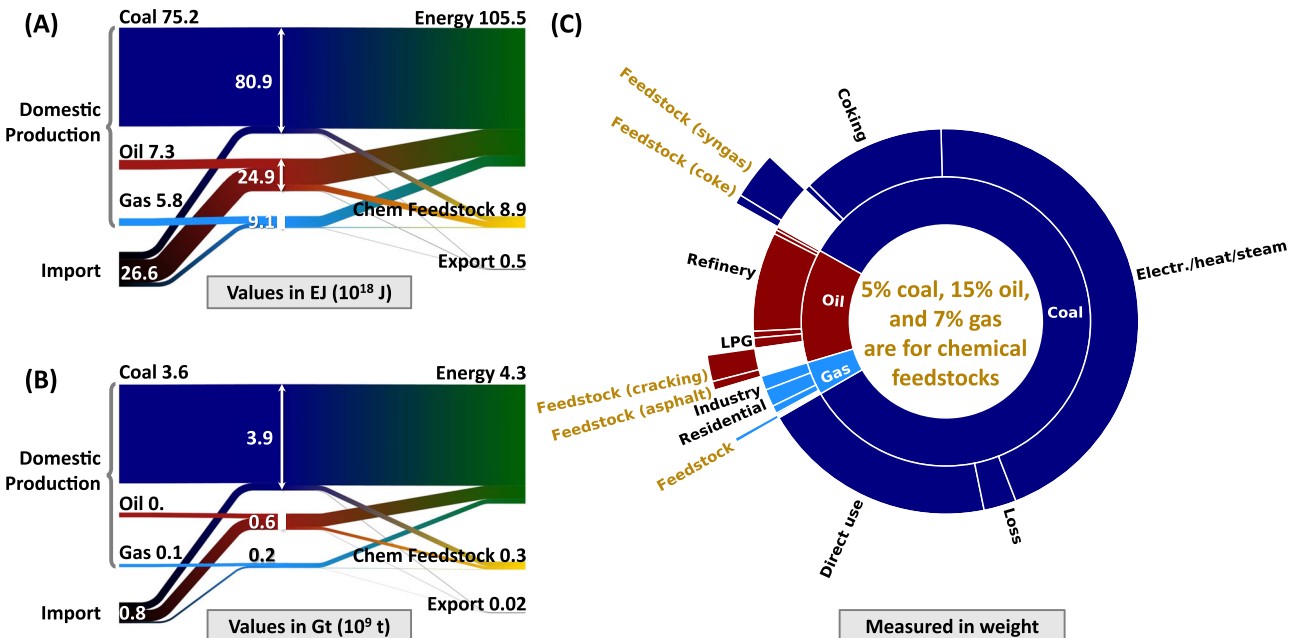

**Fig. 1 | Differentiating the use of fossil hydrocarbons as fuel and as feedstock in China. A** Material and energy flow of hydrocarbons in China, measured in units of energy (exajoule, EJ). **B** Material and energy flow of hydrocarbons in China, measured in units of mass (gigaton, Gt, which is equivalent to petagrams, Pg). **C** The pi chart depicts the use structure of fossil hydrocarbons. The inner circle explains the use structure of coal, crude oil, and natural gas. The middle circle (and the black text) describes the part related to the part used as an energy source. The outer circle (and the orange text) outlines the proportion of chemical feedstocks.

landscape. These together accounted for 92($\pm$1)% of the syngas produced. The rise of coal-based methanol-to-olefin (MTO) and methanol-to-propylene (MTP) technologies has increased the utilization of coal in the chemical sector. Our model estimated that 73.8 Mt·yr$^{-1}$ ($\pm$17%) of coal was utilized in the production of methanol, representing 43($\pm$1)% of the total feedstock use of coal and 2($\pm$0.3)% of China's total coal consumption. In 2017, MTO and MTP derived from coal contributed 20($\pm$2)% to ethylene production and 25($\pm$2)% to propylene production. Coal was also the primary feedstock for the production of ammonia. A total of 53.5 Mt·yr$^{-1}$ ($\pm$12%) of coal was used to produce syngas for the hydrogen needed to synthesize ammonia, contributing to 76%($\pm$1%) of ammonia production.

China's Polyvinyl Chloride (PVC) production is predominantly coal-based due to policy and local resources, differing from global practices. It leverages abundant coal and chlorine by-products, resulting in 76$\pm$1% of PVC being produced from calcium carbide, mainly in the coal-rich north. Conversely, the oil-based ethylene route for PVC is less common, located in coastal petrochemical complexes. Moreover, China's policy on natural gas in chemical feedstock roles[52] and prioritizes residential needs, restricting it to 11($\pm$2)% for methanol and 24($\pm$6)% for ammonia production in 2017.

Synthetic resins, particularly plastics, are the primary products derived from fossil resources in China. In 2017, China was responsible for approximately 30% of the world's plastics output, amounting to 93.9 Mt·yr$^{-1}$. This includes production volumes of 26.3 Mt·yr$^{-1}$ for polyethylene (PE), 21.9 Mt·yr$^{-1}$ for polypropylene (PP), 17.8 Mt·yr$^{-1}$ for polyvinyl chloride (PVC), 6.0 Mt·yr$^{-1}$ for polystyrene (PS), 5.3 Mt·yr$^{-1}$ for acrylonitrile butadiene styrene (ABS), and 8.4 Mt·yr$^{-1}$ for polyethylene terephthalate (PET). Coal plays a particularly crucial role in the production of PVC, accounting for three-quarters of its output, contrasting with the petrochemical origins of other plastics. The model indicates a high input-to-output ratio of fossil resources to plastics, with 0.82($\pm$18%) tons of coal and 0.64($\pm$5%) tons of crude oil needed per ton of plastic produced on average nationally (see the disaggregated results in Supplementary Data S2). Interestingly, only about 5% of the produced polymers are exported, while a substantial

volume of monomers and polymers are imported. The robust domestic demand, driven by sectors such as packaging and construction[26], alongside export-oriented manufacturing, suggests considerable potential for increased fossil hydrocarbon feedstock consumption within China in the future.

The fertilizer industry stands as the second-largest sector in chemical feedstock consumption by weight, with coal and natural gas being critical inputs. The production volume of key nitrogen fertilizers, including urea and ammonium phosphate, reached a notable 93.2($\pm$0.3%) Mt·yr$^{-1}$, with the production process heavily reliant on coal and natural gas to source hydrogen. Specifically, the production of one ton of fertilizer necessitates the use of 0.35($\pm$15%) tons of coal and 0.06($\pm$16%) tons of natural gas, highlighting the sector's substantial hydrocarbon requirements (this excludes the energy consumed for electricity, heat, and steam generation). In the international trade context, China was a net exporter of chemical fertilizers in 2017, shipping out 24.4 Mt, with nitrogen fertilizers accounting for half of this export volume. Contrastingly, potash fertilizers dominated China's fertilizer imports, totaling 7.6 Mt out of 9.0 Mt. This trade pattern underscores China's self-reliance in nitrogen fertilizer production, with minimal imports of only 0.1 Mt. The reliance on domestically produced ammonia from coal is a cornerstone for the nation's agricultural productivity and, by extension, its food security.

Moreover, the chemical industry's output for fibers and rubber totaled 44.5($\pm$0.2%) Mt·yr$^{-1}$, with each ton of product necessitating 0.25($\pm$18%) tons of coal and 0.72($\pm$1%) tons of crude oil. Additionally, the production of fuel additives and explosives amounted to 48.2($\pm$9%) Mt·yr$^{-1}$. Lastly, chemicals serving as fuels or fuel additives, such as MTBE for gasoline engines, DME, and methanol gasoline, represented a considerable volume of 32.6($\pm$12%) Mt·yr$^{-1}$, indicating a substantial portion of chemical products also contribute directly to specialized energy consumption applications.

## CO$_2$ emissions
We analyzed (gate-to-gate, i.e., scope 1 and 2) CO$_2$ emissions in chemical processes, distinguishing between direct process emissions[53]

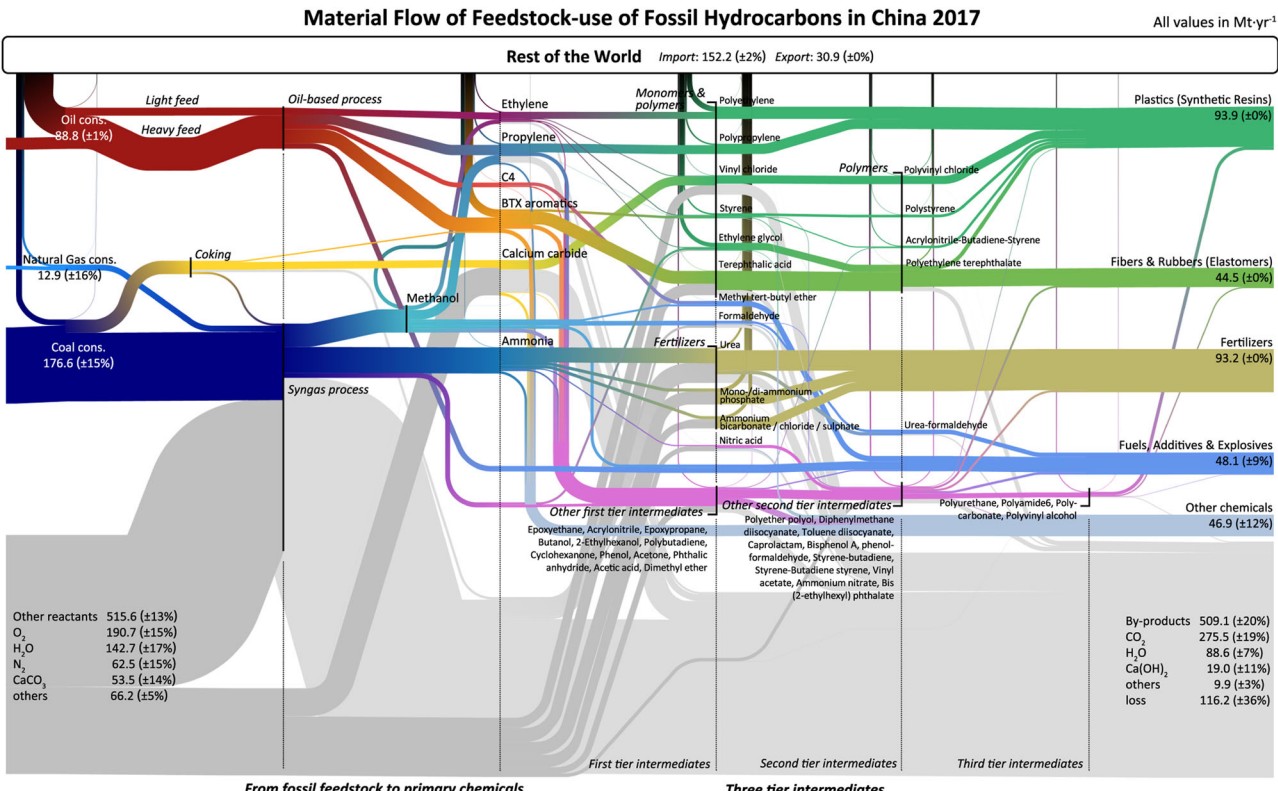

**Fig. 2 | The material flow of the fossil hydrocarbon feedstock in China.** The uncertainty ranges are noted in percentage form.

| Route | Product | Coal-based | Oil-based | Gas-based |
|---|---|---|---|---|
| Ethylene-Related | Ethylene | 20% | 80% | 0% |
| Propylene-Related | Propylene | 25% | 75% | 0% |
| C4 | Butene | 0% | 100% | 0% |
| | Butadiene | 0% | 100% | 0% |
| BTX | Benzene | 26% | 74% | 0% |
| | Toluene | 0% | 100% | 0% |
| | Ortho-Xylene | 0% | 100% | 0% |
| | Meta-Xylene | 0% | 100% | 0% |
| | Para-Xylene | 0% | 100% | 0% |
| Methanol-Related | Methanol | 89% | 0% | 11% |
| Ammonia-Related | Ammonia | 76% | 0% | 24% |
| Calcium Carbide-Related | Calcium Carbide | 100% | 0% | 0% |

**Fig. 3 | Production mix of primary chemicals from feedstocks.** The proportions are traced by the direct and indirect cost of carbon atoms embodied in the primary chemicals (embodied carbon from the feedstock). For ammonia, we track the number of carbon atoms necessary to produce hydrogen. See Supplementary Fig. S1 for results of intermediates chemicals. See Supplementary Data S1 and Supplementary Fig. S2–3 for the uncertainty ranges. See Supplementary Fig. S4 for results, including import/export.

and energy-related emissions from producing the heat, electricity, and steam used in the processes. Given primary chemicals' pivotal role in the industry, accounting for two-thirds of its total energy demand[1], we focused on their $CO_2$ byproduct. Using material balances and process-based coefficient methods, we break down the stages of chemical

processes and estimate the stage-wise carbon intensity of various chemical products, which help identify key emitting processes to target mitigation (Fig. 5 and Supplementary Fig. S9–11 for breakdowns). We determined that in 2017, process emissions from primary chemicals amounted to 0.30 Gt·yr⁻¹ (±18%) of $CO_2$, while energy-related emissions were slightly higher at 0.34 Gt·yr⁻¹ (±17%).

In examining process emissions, it becomes evident that syngas generation from coal and natural gas, along with the coal-based coking operation for producing calcium carbide, stand out as primary process emission contributors (as detailed in Fig. 2). Ammonia production, largely for hydrogen sourcing, is responsible for about half of these emissions, with coal pathways emitting 0.13 Gt·yr⁻¹ (±12%) and steam methane reforming (SMR) accounting for 22 Mt·yr⁻¹ (±12%). A third of the emissions are linked to methanol-related pathways, such as methanol-to-olefins (MTO) and methanol-to-propylene (MTP), with an output of 0.10 Gt·yr⁻¹ (±21%). The conversion of coke to calcium carbide adds another notable amount, approximately 36.5 Mt·yr⁻¹ (±14%). It's important to note that coal-fed chemical production was responsible for an overwhelming 90% of these emissions, which translates to 2.7(±0.5)% of China's national $CO_2$ emissions in 2017.

Coal's inherent low hydrogen-to-carbon (H/C) ratio and energy-per-carbon content result in higher emissions from chemical processes[40,54] than those based on oil or natural gas. To align with the H/C ratio necessary for methanol production or to obtain hydrogen for ammonia synthesis, coal-derived syngas must be processed through a water-gas shift reaction (WGSR). Specifically, for methanol synthesis, the syngas H/C ratio should be greater than 4 (H2/CO > 2). With coal's H/C ratio at approximately 0.7, additional hydrogen is generated via WGSR by converting carbon monoxide to $CO_2$. Consequently, only about 40(±2)% of the carbon in coal, when used solely as feedstock, contributes to methanol production. This limitation, rooted in coal's chemical composition, underscores the challenge of emission mitigation, necessitating alternatives to WGSR for hydrogen generation to reduce emissions. Moreover, although petroleum-based processes are

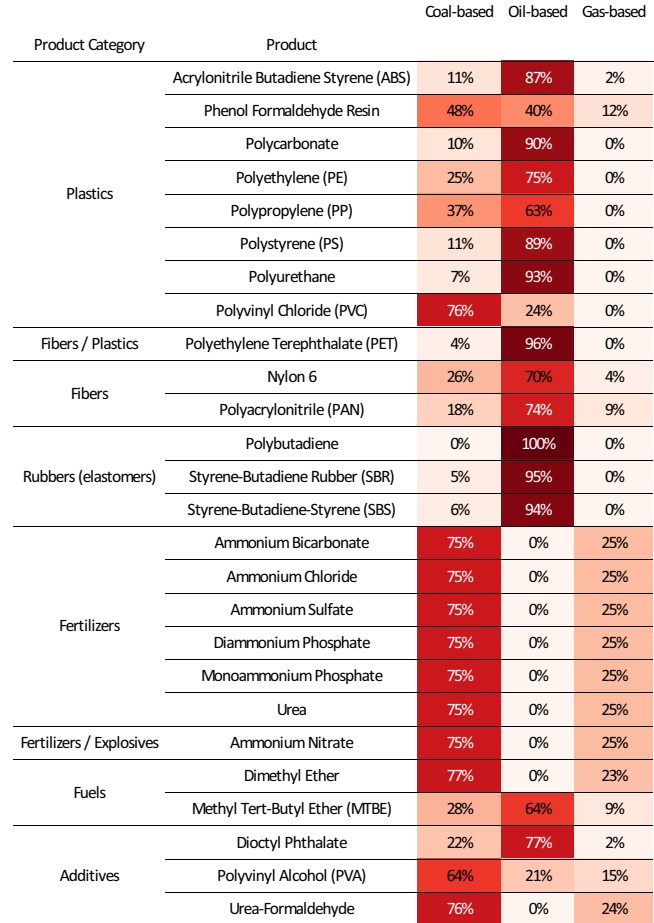

| Product Category | Product | Coal-based | Oil-based | Gas-based |
|---|---|---|---|---|
| Plastics | Acrylonitrile Butadiene Styrene (ABS) | 11% | 87% | 2% |
| | Phenol Formaldehyde Resin | 48% | 40% | 12% |
| | Polycarbonate | 10% | 90% | 0% |
| | Polyethylene (PE) | 25% | 75% | 0% |
| | Polypropylene (PP) | 37% | 63% | 0% |
| | Polystyrene (PS) | 11% | 89% | 0% |
| | Polyurethane | 7% | 93% | 0% |
| | Polyvinyl Chloride (PVC) | 76% | 24% | 0% |
| Fibers / Plastics | Polyethylene Terephthalate (PET) | 4% | 96% | 0% |
| Fibers | Nylon 6 | 26% | 70% | 4% |
| | Polyacrylonitrile (PAN) | 18% | 74% | 9% |
| Rubbers (elastomers) | Polybutadiene | 0% | 100% | 0% |
| | Styrene-Butadiene Rubber (SBR) | 5% | 95% | 0% |
| | Styrene-Butadiene-Styrene (SBS) | 6% | 94% | 0% |
| Fertilizers | Ammonium Bicarbonate | 75% | 0% | 25% |
| | Ammonium Chloride | 75% | 0% | 25% |
| | Ammonium Sulfate | 75% | 0% | 25% |
| | Diammonium Phosphate | 75% | 0% | 25% |
| | Monoammonium Phosphate | 75% | 0% | 25% |
| | Urea | 75% | 0% | 25% |
| Fertilizers / Explosives | Ammonium Nitrate | 75% | 0% | 25% |
| Fuels | Dimethyl Ether | 77% | 0% | 23% |
| | Methyl Tert-Butyl Ether (MTBE) | 28% | 64% | 9% |
| Additives | Dioctyl Phthalate | 22% | 77% | 2% |
| | Polyvinyl Alcohol (PVA) | 64% | 21% | 15% |
| | Urea-Formaldehyde | 76% | 0% | 24% |

**Fig. 4 | Production mix of chemicals (plastics, fibers, rubbers, fertilizers, explosives, fuels, and additives) from feedstocks.** The proportions are traced by the direct and indirect cost of carbon atoms embodied in the products (embodied carbon from the feedstock). For ammonia-related products, we track the number of carbon atoms necessary to produce hydrogen. See Supplementary Data S1 and Supplementary Fig. S2–3 for the uncertainty ranges. See Supplementary Fig. S5 for results including import/export.

linked to fewer direct emissions, the energy consumed in processes such as cracking and reforming still results in considerable carbon emissions.

Energy-related emissions are predominantly generated from the energy supply necessary for crude oil's cracking and reforming pathways. As shown in Fig. 5, steam cracking of ethylene is particularly emissions-intensive, with approximately 1.1 tons ($\pm 18\%$) of $CO_2$ emitted for every ton of ethylene produced and contributing 16 Mt·yr$^{-1}$ $CO_2$ emission. Additionally, the energy-intensive process of BTX (benzene, toluene, xylene) reforming contributes 1.9 tons ($\pm 18\%$) of $CO_2$ for each ton manufactured, totaling 42 Mt·yr$^{-1}$ $CO_2$. $CO_2$ emissions from coal and natural gas chemical processes are also substantial due to the high-temperature requirements of gasification and the Water-gas Shift Reaction. MTO and MTP processes together contribute to 40 Mt·yr$^{-1}$. The predominance of coal in China's energy portfolio, accounting for 55% in 2017, exacerbates these emissions, as evidenced by a comparison with other global regions in Supplementary Data S2. A prompt shift towards a more sustainable energy mix and the possible electrification of chemical processes could markedly diminish these energy-related emissions[14,18]. The extent of emissions reduction is determined by the rapidity of the system-wide energy transition and the optimization of energy efficiency, specifically within the chemical sector.

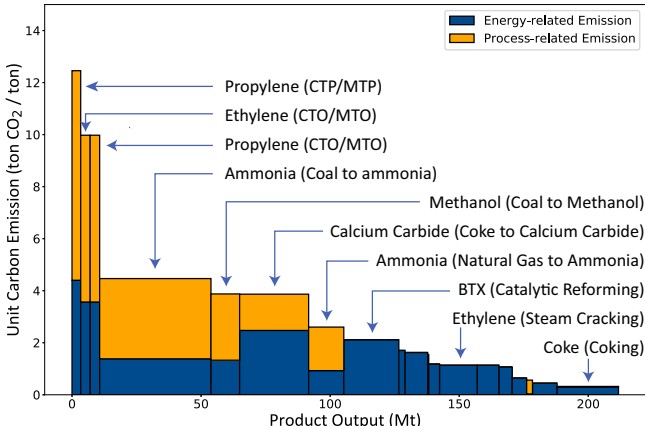

**Fig. 5 | Process- and energy-related carbon emissions of primary chemical production.** The *Y*-axis indicates the carbon emissions per unit product (ton $CO_2$ per ton products). The *X*-axis is the output of an individual product. In this form, each rectangular area denotes the total emissions of production of each product, differentiated by process-related emissions (in **yellow**) and energy-related emissions (in **blue**). Please refer to Supplementary Figs. S7–8 for the uncertainty ranges. See Supplementary Figs. S9–11 for a breakdown by product, process, and feedstock of cumulative carbon emission.

## Discussion

While the role of chemical feedstocks in the consumption of fossil fuel is often underappreciated, their increasing demand is a critical factor in the growth of global hydrocarbon use[1,14,55]. The connection between production and consumption in energy/climate scenario models has historically been obscured by insufficient attention to non-energy uses and the complexities of fossil fuel production networks, complicating the assessment of demand-side measure effectiveness. Despite the thorough exploration of supply-side measures, evidenced by many scenario analyses, they alone prove inadequate[4]. Implementing circular economy principles, demand reduction strategies, and integrated energy scenario assessments necessitates a comprehensive detail of material cycles in chemical production[56]. Levi and Cullen[4] emphasized the essential need for precise mass balances and clear documentation of the industry's primary material flows, vital for accurately analyzing the impact of mitigation strategies further along the value chain.

Reflecting on the establishment of a macro-balance within the petrochemical industry reveals several insights. Initially, mapping the network of chemical substances within a region's industrial framework is both essential and complex, going beyond merely cataloging key products and processes. It also requires an understanding of their intricate interrelations, diverging from traditional Material Flow Analysis (MFA) due to the sophisticated nature of the chemical metabolism. Insights from this work, along with previous efforts[1,4,14,57,58] could provide references, yet obtaining robust production and process data remains challenging. It often entails corroborating information from diverse sources, including statistical reports, literature, and industry studies, with regional technological diversity adding complexity to data collection, especially for smaller-scale downstream productions. Inevitably, a mass balance on this scale will be affected by inconsistencies among divergent data sources. Our methodology, incorporating a loss term to address yield discrepancies, aims to preserve the integrity of existing data while acknowledging its limitations. Such an extensive mass balance necessitates ongoing collaboration between academics and industry practitioners to improve data availability and transparency.

Engagement with industry experts was essential for validating our model's assumptions and results. The dialogue with practitioners on decarbonizing China's chemical sector revealed a difference in outlook

compared to the long-term sustainability focus of environmental experts. This difference, emphasizing immediate capabilities and economic considerations, highlights the need to challenge industry norms and incentives to encourage proactive decarbonization efforts. For instance, the Rocky Mountain Institute's projections[39] for the adoption of green hydrogen or CCS in coal-based capacities by 2030 (30%) were seen as overly optimistic by industry experts, who anticipated only a 5% integration. Moreover, continuous monitoring is needed, given the anticipated considerable capacity expansions in China and the United States, with Asia and the Middle East driving long-term growth[1]. Regional differences in process pathways, such as the reliance on ethane in North America and the Middle East versus the use of naphtha for high-value chemicals in Asia and Europe, require focused attention in modeling. These processes are major contributors to carbon emissions and play a pivotal role in downstream economics, emphasizing the need for region-specific analyses. Additionally, the petrochemical industry's dynamic nature, considering the maturity of chemical processes, necessitates regular updates to the model, with data collection, for example, every five years, to ensure its continued relevance and accuracy.

Such a mass-balance framework can extend to a simple scenario outlook as we explore China's 2030 based on industry expert estimates (see Supplementary Data S2, Notes, and Figure S12). Our findings suggest that, except ammonia, the output of most primary chemicals could double or triple from 2017 levels by 2030. Without strategic interventions, emissions might surge by 87% (+ 520 Mt) from 2017. Transitioning to a cleaner energy mix could largely lower emissions. An expected reduction in the carbon intensity of electricity from 4.88 to 4.22 tCO2/tce could decrease emissions by 80 Mt compared to maintaining the current energy mix. Furthermore, integrating green hydrogen into coal-to-chemical processes emerges as a key strategy for substantial emission reductions. A modest adoption rate of 0.5 to 5% by 2030, as forecasted by various experts, along with the application of carbon capture, utilization, and storage (CCUS) in producing olefins, methanol, and synthetic ammonia, could cut emissions by an additional 162 Mt in 2030, nearly a quarter of the 2017 levels.

This underscores green hydrogen's potential to reduce $CO_2$ emissions by up to 68% from the conventional syngas process[59]. The geographic concentration of China's coal chemical production in areas like Ningxia, Inner Mongolia, and Shaanxi, combined with a considerable renewable energy potential, suggests a promising integration of coal chemical processes, hydrogen energy, and CCUS[54]. Notably, China's relatively young manufacturing assets represent over half of the global chemical industry's capacity[39]. Such a strategic integration seems more pragmatic than allowing these young chemical infrastructures to become obsolete quickly. However, emerging technologies offer potential, and their widescale adoption by 2030 poses challenges (detailed further in Supplemental Notes 2.5–2.6).

Chinese policymakers[60] are considering measures such as enhancing energy efficiency, advancing electrification, integrating green hydrogen and CCUS, transitioning to lighter hydrocarbons, 'crude oil-to-chemical' integration, and promoting the transformation towards a circular economy, etc. Our material flow model provides detailed information, from broad process linkages to specific policy and technology intervention points[19]. However, realizing these technological advancements depends on updating existing and constructing new chemical facilities (such as green hydrogen production). Additionally, ensuring an adequate supply of low-carbon electricity is crucial for supporting electrification and CCUS efforts[18], necessitating integrated planning across the energy and chemical sectors. On the demand side, fostering changes in consumer behavior and lifestyles is vital for real demand reduction and the advancement of a circular economy, especially in altering plastic consumption patterns for substantial environmental benefits.

For future analyses, the integration of supply-side and demand-side strategies within a physics-based framework enables direct quantification of new technology adoption, considering timelines, infrastructure compatibility, investment needs, and integration ease. It also accounts for the influence of market dynamics, especially the effects of policy-driven incentives and costs on carbon emission reduction efforts. The potential simulation of recycling flows within the chemical industry is necessary but marked by ambiguity[61] that only 11–17% of the demand might be satisfied by recycled plastics. Studies are dependent on details such as exact plastic composition, contamination, and technological progress in recycling, which are difficult to capture at the macro level. Given the multifaceted nature of chemical production, focused research on plant-level emissions reduction is also practical. These insights present the potential for a more refined integration of material cycles into integrated assessment models (IAMs)[56], thereby enriching climate change and fossil hydrocarbon scenario discussions. This holistic approach underscores the need for continuous model monitoring, data-driven scientific support[62], and proactive policymaker and practitioner engagement to enhance carbon emission management, facilitate carbon data accessibility, and guide enterprises towards energy-efficient and low-carbon practices.

## Methods

The study models fossil hydrocarbon processing in three stages: total consumption split to fuel and feedstock (blue), conversion from fossil feedstock to primary chemicals (yellow), and further manufacturing to downstream products (green), as shown in Fig. 6. Our primary focus is on the yellow and green sections, emphasizing fossil hydrocarbon feedstock use.

Following is the overall workflow and principles of this study:

(1) Determine total fossil feedstock inputs (yellow). There are 21 processes for producing 7 primary chemicals (ethylene, propylene, C4 olefins, BTX aromatics, ammonia, methanol, and calcium carbide), and 5 processes directly utilize fossil hydrocarbons as feedstock. We categorize these 26 processes into three distinct groups and determine the total fossil feedstock inputs based on the production yield of each chemical and the process-based coefficients and stoichiometry.

(2) Split energy and feedstock use of hydrocarbons (blue). This is addressed through a mass balance approach, based on the result from the previous step and energy-balancing data from the *China Energy Statistics Yearbook*[63].

(3) Connect primary chemicals with downstream production (green). There are 48 chemicals and 51 processes for the three tiers of downstream production. In this model, each production process is treated as a node, where mass balance relationships for reactants and products are established based on the production yield of each chemical and the process-based coefficients and stoichiometry. See Figure S13–18 for their chemical interlinks. Acknowledging the inherent yield inaccuracies in macroscopic data modeling, we do not use an enforced balance like harmonization techniques in STAN[64]. To be specific, for inconsistencies, we assess the quality and reliability of the data, including expert consultations, to determine whether input or output data is more dependable. Subsequently, we take the remaining values into imbalance items, which are documented alongside other 'losses' to maintain data integrity while recognizing its inherent limitations.

(4) Trace production mix and feedstock intensity (yellow/green). Based on the interwoven material flow results from the previous steps, we can trace the detailed production mix (i.e., the proportion of three production processes, coal-based, oil-based, or gas-based, of the total domestic production amount) and feedstock intensity (the average amount of fossil feedstock used per ton of product) of each chemical. These values build quantitative bridges between the final products and fossil feedstocks.

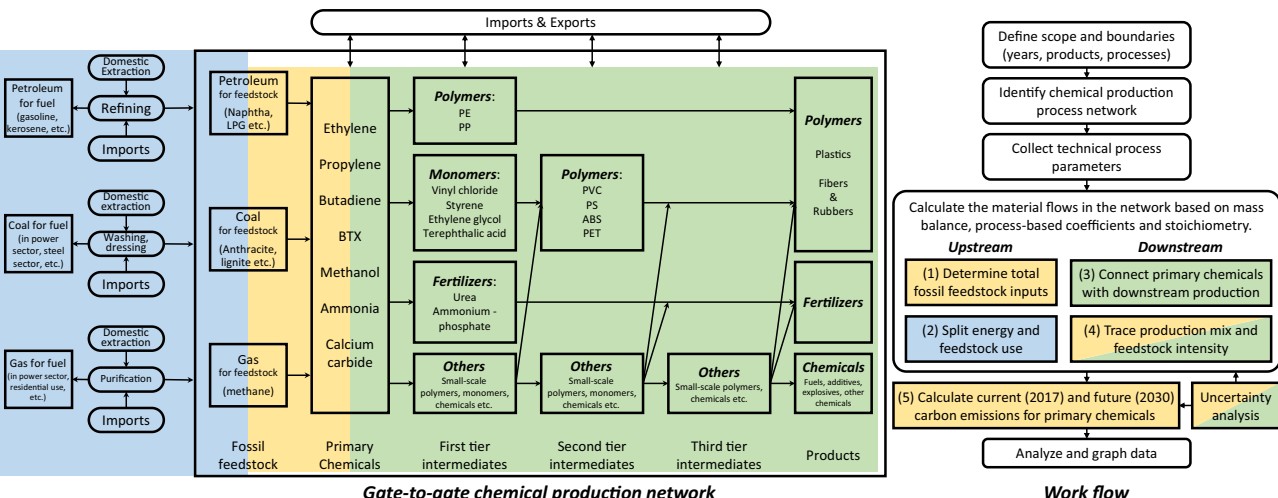

**Fig. 6 | System boundary and workflow of this study.** The diagram uses three colors to represent different stages of the modeling: total fossil fuel consumption (**blue**), primary chemical production (**yellow**), and downstream chemical production (**green**).

(5) Calculate carbon emissions for primary chemical production processes in 2017 and quantify future mitigation potential in 2030 (yellow). We calculate the scope of two carbon emissions of the 21 primary chemical production processes in China, in 2017. The process-related emission (or direct emission) is the $CO_2$ as the by-product of the chemical reaction. The energy-related emission (or indirect emission) is the equivalent $CO_2$ of the electricity and heat consumption in the production process. We further conduct a simplified scenario analysis to quantify the short-term mitigation potential for China's primary chemical production in 2030. Three distinct scenarios include the baseline scenario (only upscaling the chemical demand), grid improvement scenario (enhanced energy structure), and technology scenario (adoption of low-carbon technologies).

(6) Uncertainty analysis (yellow/green). We apply Monte Carlo simulations to evaluate the uncertainties in our material flow model. We categorize the process-based coefficients into four types and assign different levels of uncertainties. We arrive at the final confidence value by running 20,000 rounds.

All parts will be detailed below, and further details are noted in Supplemental Notes 2.1–2.6 and Data S2.

### Determine total fossil feedstock inputs

Steam cracking and other oil-based processes. Steam cracking is identified as the primary method for producing ethylene, propylene, C4 olefins, and BTX aromatics. Approximately 80% of oil-based ethylene is produced through steam cracking, which also simultaneously yields propylene, C4 olefins, BTX aromatics, and by-products like $CH_4$. For this study, average feedstock input and product ratio data for steam crackers in China were sourced from factory estimates (detailed in Data S2). The total feedstock inputs, primary chemicals, and by-product outputs were calculated based on the ethylene yield. Carbon emissions from these multi-product processes were allocated by mass[65], following the same mass allocation method used in the refinery process, where the Share of Mass Content (SMC) of each product is determined for allocation purposes. The model also incorporates other oil-based processes such as catalytic cracking, reforming, and dehydrogenation.

Syngas process and methanol-to-olefin. The syngas process, integral for producing ammonia and methanol, comprises two stages: (i) syngas preparation and (ii) chemical synthesis. In the preparation stage, crude syngas are generated from coal via the Partial Oxidation Process (POX) and from natural gas through the Steam Reforming Process (SMR). Subsequent adjustment of the CO/H₂ ratio is achieved through the Water-Gas-Shift Reaction (WGSR) to meet specific production requirements. The subsequent chemical synthesis stage maintains a consistent reaction mechanism regardless of the fossil feedstock, requiring a CO/H₂ ratio of 1:2 for methanol production. For ammonia, it is presumed all CO converts to $CO_2$, optimizing H₂ yield for mass balance purposes. Additionally, the model incorporates the Methanol to Olefins (MTO) and Methanol to Propylene (MTP) processes, which convert approximately 50% of methanol into ethylene and propylene. We also include the production of ethylene glycol, hydrogen (used in cyclohexanone, methylene diphenyl diisocyanate, and toluene diisocyanate synthesis), carbon monoxide (for acetic acid production), and syngas (for 1-butanol and 2-ethyl hexanol).

Coke, calcium carbide, and vinyl acetate production. (1) The coking process yields coke for calcium carbide and coke oven gas for methanol and benzene. (2) Calcium carbide is produced by calcining limestone and coke, subsequently reacting with water to release acetylene gas. Vinyl chloride monomer, a precursor to PVC, is synthesized through the addition reaction of acetylene with hydrogen chloride, followed by polymerization. This process is particularly prevalent in China and dominates the calcium carbide consumption. (3) Vinyl acetate, a downstream chemical, is synthesized through the addition reaction of acetylene with acetic acid. The acetylene used to produce vinyl acetate could be sourced from calcium carbide, methane, or ethylene. Only the methane-based process is included here to estimate the natural gas feedstock usage, while the other two are categorized as a downstream process.

### Split energy and feedstock use of hydrocarbons

Based on the above results, we split the use of fossil hydrocarbons for both energy and feedstock purposes. Coal's downstream applications include conversion (inclusive of feedstock use), direct consumption, export, and losses. Crude oil is employed in refining and chemicals, direct consumption, export, and loss. Natural gas fulfills diverse roles, serving as industrial fuel, residential heating, power generation, and as a feedstock in the chemical industry. In our analysis, we distinctly separate the portion used as feedstock from the general consumption pattern. Figure 1 illustrates this division of fossil hydrocarbons into fuel and feedstock categories. The energy use of fossil hydrocarbons can be found in the energy balance sheet of *China's Energy Statistical Yearbook*[45]. Details of split and mass balance are shown in sheet "2 Split for hydrocarbons" in Supplementary Data_S2_Flows.xlsx.

## Connect Primary Chemicals with Downstream Production

Based on the actual production process, we divided the downstream chemicals into three tiers, with a total of 48 kinds. Generally, if a chemical had a yield larger than 1Mt in 2017, it was selected as a "node chemical". Certain adjustments were made to ensure the integrity of the production process. Among them, 24 chemicals are in the first tier, 19 chemicals in the second tier, and five chemicals in the third tier. In the Sankey diagram, each node represents the production process of a chemical, and the process of connecting the nodes is similar, as described below.

Basic concepts. In the calculation for a special node chemical with the ordinal number $i$, 'Upstream chemical' is a node chemical (which can be numbered as $i-1$) to produce node chemical $i$. 'Downstream chemical' refers to a node chemical (which can be numbered as $i+1$) that consumes node chemical $i$ for production. 'Other reactants' include $O_2$, $N_2$, $HCl$, etc., that are also needed for production. 'By-products' refers to $H_2O$, $CO_2$, etc., produced along with the node chemical. 'Losses' is the difference between the mass of (upstream chemical + other reactants) and (downstream chemical + by-products), which also consists of unbalanced terms.

Basic Data Required. Generally, the required data are:
(1) the yield of node chemical $i$ - $Y_i$
(2) the import and export volume of node chemical $i$ - $Imp_i$ and $Exp_i$
(3) the main processes for producing node chemical $i$ and their corresponding chemical reaction equations – each process with an ordinal number $j$
(4) the share of processes $j$ for producing the $i$ node chemical – $R_{ij}$
(5) the process-based coefficients for process $j$ to produce node chemical $i$ - $Q_{ij}$
(6) the consumption of upstream chemicals – $C_{i-1}$
(7) the proportion of upstream chemical used to produce node chemical $i$ through process $j$ - $P_{i-1,j}$

Calculate Inflows of a Node. Inflows for each node include:
(1) upstream chemical inputs of each process to produce node chemical $i$ - $Inp_i$

$$Inp_i = \sum_j Y_i \cdot R_{ij} \cdot Q_{ij} \qquad (eq1)$$

Equation 2 (eq2) represents a bottom-up method to link different nodes. If $R_{ij}$ is not available, use the following eq3, which represents a top-down method:

$$Inp_{ij} = C_{i-1} \cdot P_{i-1,j} \qquad (eq2)$$

(2) other reactants inputs of each process to produce node chemical $i$ - $OthInp_i$. $OthInp_i$ can be obtained mainly by stoichiometry calculation.
(3) import volume of node chemical $i$ - $Imp_i$

Calculate Outflows of a Node. Outflows for each node include:
(1) the amount of node chemical $i$ to produce downstream chemical - $Outp_i$
$Outp_i$ shall be equal to $\sum Inp_{i+1}$ due to mass balance, which can be calculated by the same method as described in the previous paragraph.
(2) by-products output of each process to produce node chemical $i$ – $OthOutp_i$
$OthOutp_i$ can be obtained mainly by stoichiometry calculation.
(3) the loss of each process to produce node chemical $i$ - $L_i$

$$L_i = Inp_i + OthInp_i - Outp_i - OthOutp_i \qquad (eq3)$$

(4) export volume of node chemical $i$ - $Exp_i$

(5) others - $Oth_i$

$$Oth_i = Inp_i + OthInp_i + Imp_i - Outp_i - OthOutp_i - Exp_i \qquad (eq4)$$

Eq4 reflects the part of node chemical $i$ that is not involved in the subsequent production process, which is represented in the Sankey diagram as "other chemicals". The unbalanced terms are included in 'losses'. All the process data and adjustments above are shown in Supplementary Notes and Supplementary Data S2.

## Trace Production Mix and Feedstock Intensity

Production mix ($PM_{coal/oil/gas,i}$ %) is the proportion of three production processes (coal-based, oil-based, or gas-based) of the total domestic production amount of each chemical. It is directly related to the share of processes $j$ for producing the $i$ node chemical ($R_{ij}$).

(1) For primary chemicals, domestic production determines their $CO_2$ emissions (not GHG emissions) and sources. Each of the $j$ processes for producing each chemical is either coal, oil, or gas-based. Thus, $PM_{coal/oil/gas,i}$ is the sum of relevant $R_{ij}$.

$$PM_{coal,primarychemical} = \sum_j R_{ij,coal-relevant} \qquad (eq5)$$

(2) For downstream chemicals, however, since this model is for a single country (China), the imports and exports shall be taken into consideration. Thus, we define the Consumption Mix ($CM_{coal/oil/gas,i}$) of primary chemicals, which is the proportion of the three production processes (coal-based, oil-based, or gas-based) of the total consumption amount (consumption = domestic production + import-export) of each chemical. Here we assume the imported primary chemicals are all oil-based (olefins and aromatics) or gas-based (ammonia and methanol) since there is no calcium carbide import in China, 2017. The $PM_{coal/oil/gas,i}$ for the downstream chemical $i$ is obtained as a weighted average of $CM_{coal/oil/gas,i}$ for upstream input chemical $i-1$.

$$PM_{coal,i} = \sum W_{i-1} \cdot CM_{coal,i-1} \qquad (eq6)$$

The feedstock intensity ($FI_{coal/oil/gas}$, t/t) is the average amount of fossil feedstock used per ton of product. The feedstock intensities of eight synthetic resins are calculated in this study to estimate the fossil feedstock reduction potential of recycled plastic pellets.

$$FI_{coal,primarychemical} = \sum_j R_{ij} \cdot Q_{ij,coal} \qquad (eq7)$$

$$FI_{coal,i} = \sum_j R_{ij} \cdot FI_{coal,i-1} \qquad (eq8)$$

Eq6-eq9 are examples of calculating coal-based processes. See Supplementary Notes and Supplementary Data S2 for detailed information.

## Calculate carbon emissions of primary chemical production processes

We identified the 21 primary chemical production processes using the coefficient-based carbon accounting method to evaluate the scope 2 carbon emissions. To be clear on scope, we define them as:

(1) *Carbon emission* is the direct and indirect $CO_2$ emitted from the production process. We first calculate the carbon intensity (tCO$_2$/ton product) and then the total emissions (tCO$_2$). (2) *Process-related carbon emission* (direct emission) is the $CO_2$ as the by-product of the chemical reaction where fossil hydrocarbons are converted to chemicals. (3) *Energy-related carbon emission* (indirect emission) refers to

the indirect carbon emission related to electricity, heat, etc., in the production process.

The direct carbon intensity is calculated from the stoichiometry of each reaction process, following the mass balance. The indirect carbon emission intensity ($CO_{2,ind}$, tCO$_2$/t) is calculated as follows (detailed description in Supplemental Notes):

$$CO_{2,ind} = \sum_{j=e,h} Q_j \cdot E_j \qquad (eq9)$$

$Q_j$ - the emission factor of energy (electricity, heat, etc.) (t CO$_2$/tce); $E_j$ - the energy consumption intensity of primary chemicals production processes (tce/t product).

Two examples of the carbon accounting of primary chemical production processes can be found in Supplementary Notes 2.3. The emission factor and energy consumption intensity are listed in Supplementary Data_S2 sheet 1.1.

It should be also noted that hydrogen (H$_2$), a key feedstock in the chemical industry, is primarily derived from fossil hydrocarbons via the syngas process and water gas conversion reaction. However, as most oil-based hydrogen is used in the hydrogenation process of refining – a component outside our model's scope – this study does not specifically include oil-based hydrogen production processes.

## Uncertainty analysis

Refineries and chemical plants, even in the same area, often have different process parameters. Therefore, an uncertainty analysis was used to determine the potential deviation of results. Here we used Monte Carlo simulations to evaluate the uncertainties in our material flow model[46,66]. In our MFA modeling process, there are mainly two types of data inputs: output of the products (yield data) and process-based coefficient data (product/feedstock ratio, etc.). We assumed the yield data (from the official yearbook[44,45]) are relatively accurate, and we focused on the uncertainties of process coefficients. All uncertain coefficients and parameters in the Monte Carlo simulation are described by normally distributed independent random variables due to the limits of process data samples. We assumed the baseline value is the initial guess—setting it as the mean of normal distribution. The uncertainty range of each individual data point was given based on the quality of the sources and the suggested uncertainty range obtained from factory data, literature, and experts' experiences (see Supplementary Notes and Supplementary Data S2 for details). Further, we used [1/6*deviation range] (for example, 0.4/6 for a deviation range of ±20%) as the standard deviation in this approach to cover 99.7% of the range of change (within 3 standard deviations in a normal distribution). Finally, we combined different uncertainties to arrive at the final confidence value by running 20,000 Monte Carlo Simulations. We chose the 95% certainty level where the lower-bound and upper-bound are noted as 2.5% and 97.5% percentiles[64]. We do not indicate the uncertainty range for data sourced from Statistical Yearbooks and results with a deviation interval of less than 0.5%.

## Quantify future emission reduction potential

Since the primary goal of this study is to present a holistic material flow diagram linked to carbon emissions in China, our scenario analysis is a simplified experiment to quantify short-term emission potential, setting the stage for more extensive future evaluations. We analyzed current production forecasts and the technical architecture for China's 2030 primary chemicals using estimates from top institutions[39,67–72], petrochemical experts, and literature[2,59,73–78]. Based on this, we constructed a simplified calculation for 2030. Within our framework, we project the production of primary chemicals, estimate and parametrize the 2030 market adoption of new technologies (including

feedstock mix change, green hydrogen coupling, direct olefin production from syngas, etc.), and calculate the overall emission intensity by considering both energy (including green electricity, etc.) and process-related emission intensities. We present three distinct scenarios for comparison: (1) In the Baseline Scenario, production volumes are projected to rise without any changes in intensity factors or advancements in technology. (2) Grid Improvement Scenario envisions an enhanced energy structure aimed at reducing the grid's emission intensity. (3) Building on this, the Technology Scenario proposes the adoption of low-carbon technologies, such as green hydrogen, further augmenting the progress made in the Grid Improvement Scenario.

For detailed descriptions, parameters, and results, see Supplementary Notes and Supplementary Data S2 (*Sheet 5 CO$_2$ mitigation potential*). Since routes involving biomass and CO$_2$ catalytic hydrogenation are expected to undergo commercialization or be put into production in actual factories post-2030[76], they are not included in this analysis. CCUS, with a potential for 50 Mt emission reduction and subject to industry debate due to external factors[75], is considered in aggregate rather than by distinct technology. Recycling and a circular economy could diminish the demand for raw materials. Given our framework directly estimates the primary production volumes, it does not include the recycling flows. Other technologies are expected to see widespread use after 2030, and their potential emission reductions are also supplemented in Supplementary Data S2.

## Data availability

The flow data (Supplementary Data S2) generated in this study have been deposited in the Zenodo database under the accession code https://doi.org/10.5281/zenodo.10836152. The Yield data is primarily sourced from the China Chemical Industry Yearbook and China Energy Statistics Yearbook[44], supplemented by literature, industry reports, and online materials. Import/export data collected from the online Customs statistics system[79]. Quality of process-based data is ensured through cross-validation using various sources, including factory data, environmental assessment reports, literature, industry analysis reports, authors' industry experience, and expert consultation. Source data are provided in this paper (Supplementary Data S1). Further requests for resources should be directed to and will be fulfilled by the corresponding authors.

## Code availability

The code to process the Monte Carlo simulation is available upon request and can be used to reproduce the results of this study.

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

## Acknowledgements

This work was supported by the Tsinghua-Sinopec Joint Institute for Green Chemical Industry Supporting Project (no. 421126 (B.Z., D.C.)) and the CIRCOMOD project (funded by the Horizon Europe research and innovation program under grant agreement no. 101056868 (E.H.)). The authors thank Prof. Qiang Zhang of the Department of Chemical Engineering and Prof. Jinping Tian of the School of Environment at Tsinghua University as well as experts we consulted from Sinopec for their suggestions.

## Author contributions

B.Z., E.H., M.J. and Y.C. conceived the research. M.J. and Y.C. developed the material flow model. M.J., Y.C., C.L., H.Y. and J.J. contributed to methodology development and data collection. M.J., Y.C., C.L., B.Z., E.H., D.C., W.Z., Q.W., H.Y., Y.R. and S.H. analyzed the results. M.J., Y.C., B.Z., E.H., D.C. and W.Z. wrote the paper. All authors (M.J., Y.C., C.L., D.C., W.Z., Q.W., H.Y., J.J., Y.R., S.H., E.H. and B.Z.) were involved in the discussions and approved the manuscript. M.J. and Y.C. contributed equally.

## Competing interests

The authors declare no competing interests. The views presented in this publication are solely those of the authors and should not be interpreted as representing the views or positions of any organization.
