## [Peer Review File · Nature Communications]

Tracing Fossil-Based Plastics, Chemicals and Fertilizers Production in ChinaEditorial Note: Parts of this Peer Review File have been redacted as indicated to remove third-party material where no permission to publish could be obtained.

Reviewers' Comments:

Reviewer #1:

Remarks to the Author:

The manuscript presents a systemic model on the structure, feedstock, processes and emissions of chemical production in China. The topic and results are interesting and relevant in the field, especially since chemical production in China is unique due to the large-scale application of coal gasification and the general lack of transparent data and investigations in this globally relevant sector of the Chinese economy. The results in the presented form therefore already justify a publication.

The applied methodology and data are sound for the level of detail that is addressed in the investigation and described in appropriate detail. The quality of process balance data (type of input data, source) should be communicated more clearly in an earlier part of the manuscript. If the majority of the applied numbers are taken from a single source, the source should be described in the introduction or before presenting results.

However, I would not recommend to publish the manuscript in the current form. Possible pathways to decrease greenhouse gas emissions (transition to renewable energy production, CCUS, green hydrogen-coal, chemical recycling) are only described qualitatively and do not contribute to the significance of the manuscript. The presented systemic model should enable the authors to integrate alternative process balances and quantitatively describe the impact of application of these technologies. Thereby, a perspective for emission reduction of the Chinese chemical industry via different pathways can be shown, which would be the most relevant and interesting result of the investigation.

Following some detailed remarks:

L108 and following: No need to summarize the results in the introduction.

P7: Figure 3 and Figure 4 do not contribute to the understanding and can be left out, since detailed data is provided in the supplementary.

L313: Missing graphs on cumulated emissions with fractional distributions to products, processes (e.g. cumulated emissions from steam cracking, coal gasification) and feedstock (naphtha/natural gas/coal)
Question on the uncertainty analysis methodology: Generally, process product yields and CO₂ emissions are associated (e.g. for coal-to-methanol production. Was this considered?

The following publications could be helpful:

<https://doi.org/10.3390/en13184859>

<https://doi.org/10.1016/j.resconrec.2021.106106>

<https://doi.org/10.1016/j.dib.2022.107848>

<https://doi.org/10.1016/j.egypro.2017.03.1821>

Reviewer #2:

Remarks to the Author:

The novelty, hypothesis of the methods, and the main assumptions of the models still need further clarification. Certain issues require addressing before the paper can be accepted for publication.

Title

1. This work is mainly focused on tracing the non-energy use of fossil fuels and carbon emissions, including plastics, fertilizers, and chemicals as the author mentioned in the main context, not including energy. So it seems that the title is not consistent with what is covered in the main body text.

Abstract

2. Line 27-30, "Results show China utilizes 5% of the consumed coal, 15% of crude oil, and 7% of natural gas as chemical feedstocks" please indicate the base year and the base value (in energy basis or mass basis) of the quantitative results. Also please ensure consistency throughout the context, e.g.,

page 3 lines 109-111 and line 145-149.

3. Line 29, "Coal heavy processes including methanol, ammonia, and PVC plastics production contribute to ~3% of China's national carbon emissions"---this information is not found in the Results and the Discussion sections, there should be somewhere to explain and discuss this point. Otherwise, it should be presented in the Abstract. Also, since this work is based on gate-to-gate LCA, rather than cradle-to-grave LCA, the rationale of the percentage statement is worth debating.

Introduction

4. Line 101, "Chemicals, plastic, and fertilizer production contribute to 10-14% of China's total emissions, equivalent to" Please provide the base year of these values.

5. The novelty needs further explanation. Is this the first work to trace fossil-based chemical production? If so, is it worth mentioning, or should explain the novelty in comparison with previous works?

Results

Fuel use and feedstock use

6. Line 125-130, Figure 1, mentioned "In 2017, China consumed 4.7 Gt of fossil hydrocarbons, including 3.9 Gt/yr of coal, 0.6 Gt of crude oil, and 0.2 Gt of natural gas", and "In 2017, 93% of the coal demand was met by domestic production while 70% of crude oil and 37% of natural gas were imported to meet the domestic demand." are these figures calculated or quoted statistics? If they are quoted figures, please provide data sources.

7. Line 142, "using material flow modeling, we are able to distinguish between the two constituents of fossil hydrocarbon consumption: fuels for energy production and feedstocks for chemical production"---the sentence is confusing: are the overall fossil consumption, energy production, and feedstocks for chemical production quantified in this work?

8. Data in Figure 2 is not consistent with the figure in lines 184-186.

9. Line 144, "Figure 2 depicts a comprehensive utilization structure of fossil hydrocarbons in China. The results show that, Consequently, the chemical industry utilize....as feedstocks". ---the sentence is confusing, do these results have anything to do with Figure 2? I suppose they should be illustrated in Figure 1. However, even so, it is still unclear how these numbers came out. And on what basis? Energy or mass basis?

Mapping Chemical Production Flows

10. I would suggest unifying the units in Figures 1 and 2, all on an energy basis or on a mass basis to avoid confusion.

11. Lines 184-185 and Line 191, the figures are not consistent with those in Figure 2.

12. Please improve the quality of Figure 2-4.

CO2 Emission

13. Line 273. In the first paragraph of this section, maybe it is necessary to add one or two sentences to explain the LCA methods (cradle-to-grave or gate-to-gate system boundary) and the selected functional units.

14. Repeat information from lines 283-285 and lines 320-322. If not, please clarify more clearly.

15. In the abstract, the author mentioned "Coal-heavy processes contribute to 3% of China's national carbon emissions", this information should be explained and discussed in the results section.

16. The carbon intensity of different chemical products are quantified as the author mentioned, then it might be helpful to illustrate the stage-wised carbon intensity of different chemical product to identify the key emission processes.

17. I would suggest to compare the carbon intensity of China's chemical products and the European's or other places.

Discussion

18. Limitations should be discussed.

Methods

The method section needs further explanation:

19. In your mass-balance model, how did you trace the use of different types of fossil fuels for different chemical products? Did you get the chemical product yield (in the year 2017) first, and then reverse the raw material (coal, oil, and gas) consumption based on mass balances? If so, then how did you assume and rationalize the assumption of the proportion of different raw material production

routes? I think it would be more clear if the author could explain the overall idea of the methods and the key assumptions first, and then go into detail to explain the mass balances of different routes.

20. In your LCA-based carbon emission calculation models. Please clarify the system boundary, functional units, and allocation basis (energy-based, mass-based, or economic-based, and why so?) should be explained. Key assumptions should be listed or explained.

Supplemental material

21. On page 15, the calculation is confusing. It seems the functional unit of this calculation is tCO₂/t Naphtha (the input raw material), rather than tCO₂/t product.

22. Supplementary note Page 14, the electricity emission factor seems too high for China's coal-fired fleets.

Reviewer #3:

Remarks to the Author:

The manuscript submitted by Jiang et al. investigates the use of fossil fuels for the production of chemicals in China. The analysis includes mapping the fossil fuels feedstock in the chemical industry and the computation of the associated scope 1 (process) and scope 2 (energy input) CO₂ emissions. Authors investigate both the production of primary chemicals (e.g. methanol) and of final chemical products (e.g. polyethylene). Generally speaking, reconstructing the carbon cycle of the chemical industry, especially when done for a diverse and large country as China, is a complex task that requires a clear and smart methodology, and a large number of input data from scientific, industrial, and institutional literature. Per se, the topic would fit well the journal aim and scope, and would be highly relevant for a large scientific community and for several policy makers. Generally speaking, the manuscript is clearly written (English is ok) and to the point, but it misses a compelling underlying narrative. It is for example unclear if authors aim at providing a picture of the carbon flow in the chemical industry, if they want to discuss possible solutions in a world of constrained fossil fuels, or both. Figures and plots are of good quality and help the understanding of the paper.

However, in the reviewer's opinion, the paper does not reach the quality for publication in Nature Communication. There are several aspects that support this conclusion.

First, the analysis of the results is very shallow; it is unclear what the main findings and take-home messages are. The paper does not provide any conclusion, thus leaving the reader with a feeling of unfinished work. In the text, authors mostly report numbers presented in the figures without a (or a very limited) critical analysis of them. They also try to identify solutions for the reduction of scope 1 and 2 CO₂ emissions (biomass, CCUS,..), but they do that in a un-organized, qualitative way spread across different paragraphs of the results and discussion. For example, it is rather disappointing to read at line 382 of page 11 that authors identify CCUS as a negative emissions technology, which is indeed wrong unless applied to biomass-derived CO₂ and associated to permanent carbon storage. For publication in this journal, authors should provide a more insightful and detailed analysis.

Second, the method and associated data input are very unclear. While authors provide a substantial supplementary information file and data files, it remains almost impossible to judge if the numbers presented in the paper are likely to be correct or not. While I do recognize that similar analyses, i.e. closing the carbon cycle of a national chemical industry, is very difficult and almost impossible to prove, I believe authors should have done a much better job in illustrating and explaining the method, which remains shallow and unclear in both the manuscript and the SI. Moreover, authors use several (grey) sources for their analysis, for example in the second excel data file, that are difficult to trace online or unspecified. For instance, I tried to look for ref.1 (China Petroleum and Chemical Industry Federation, China Chemical Industry Yearbook 2017-2018. China Chemical Information Center: Beijing, 2019) but could only find the CPCIF annual report (2019) which did not contain the data shown in the excel file. Ref. 5 in the same excel file is not a real reference: it namely says 'Customs statistics online search platform'. Authors should make the original data sources available to the reader, not only the data itself. As for the method, I find the explanation of the method rather poor and superficial. Authors could start their revision with a block diagram containing the different steps of

the method. Another disappointing point concerns the fact that in the SI authors refer to an Experimental procedures section, which however I could not find in any of the files sent with the submission. For example, authors mention that they have done a Monte Carlo simulation to tackle uncertainties, but there is no explanation of it (they again cite an Experimental procedures section which cannot be found).

To conclude, I find the paper and the presented analysis of interest for the journal and worth considering for publication; however, the scientific quality of the submitted manuscript and the associated supplementary information and data files do not reach the minimum standard required for a scientific journal.

Response to Reviewer #1:

(1) The manuscript presents a systemic model on the structure, feedstock, processes and emissions of chemical production in China. The topic and results are interesting and relevant in the field, especially since chemical production in China is unique due to the large-scale application of coal gasification and the general lack of transparent data and investigations in this globally relevant sector of the Chinese economy. The results in the presented form therefore already justify a publication.

Response:

Thank you for your remarks about the quality of the study and your suggestions for improving the manuscript. We have carefully revised it following your comments below.

(2) The applied methodology and data are sound for the level of detail that is addressed in the investigation and described in appropriate detail. The quality of process balance data (type of input data, source) should be communicated more clearly in an earlier part of the manuscript. If the majority of the applied numbers are taken from a single source, the source should be described in the introduction or before presenting results.

Response:

Thank you for your suggestions. Following them, we have revised the paragraphs in the Introduction Section to communicate more clearly regarding the data collecting and processing. The updated section in the Introduction now reads:

In this study, we present a tailored process-based material flow model for China's chemical industry, highlighting the interwoven mass and carbon flows within the expansive (gate-to-gate) production network. While global chemical flows have been mapped by Levi and Cullen [4], our research uniquely focuses on China's chemical industry, which stands out as the world's largest and is heavily reliant on coal. We further assess the process emissions arising directly from chemical production, alongside energy emissions stemming from the provision of heat, electricity, and steam by using the coefficient-based carbon accounting method (scope 2 accounting). This allows us to pinpoint the production of primary chemical building blocks and identify the most carbon-intensive processes.

Collecting foundational data for mass and carbon flow operations within the hydrocarbon-based chemical industry is challenging [4,41]. This study seeks to overcome these data limitations by sourcing and cross-verifying information from various resources, including the yearbook (China Chemical Industry Yearbook [42], Energy Statistics Yearbook [43]), disclosed industrial reports, other databases, and engagements with industry practitioners, all synthesized with an underpinning of chemical engineering knowledge. We also use Monte Carlo simulation [44] to explore potential deviations. All results presented in this study are based on data from the year 2017.

Moreover, for greater clarity and to facilitate data verification, we have

significantly enhanced the *Supplementary Data S2* by including titles of data sources in Chinese, their English translations, and direct access links. We have revalidated every data link. We also mark each link according to its accessibility status (e.g., open access, subscription required, or private).

(3) However, I would not recommend to publish the manuscript in the current form. Possible pathways to decrease greenhouse gas emissions (transition to renewable energy production, CCUS, green hydrogen-coal, chemical recycling) are only described qualitatively and do not contribute to the significance of the manuscript. The presented systemic model should enable the authors to integrate alternative process balances and quantitatively describe the impact of application of these technologies. Thereby, a perspective for emission reduction of the Chinese chemical industry via different pathways can be shown, which would be the most relevant and interesting result of the investigation.

Response:

Thank you for your constructive suggestions.

We agree and acknowledge that a quantitative analysis that explores and quantifies potential emission reduction trajectories within the Chinese chemical industry, offers broader implications. The central aim of our study is to provide a comprehensive material flow landscape linked to key carbon emissions processes in chemical production.

To address this, and guided by your suggestions, we have added a focused experimental analysis to provide a preliminary quantification of the potential emission reductions in the short term (in 2030, the target year for China's overall carbon peak), setting the groundwork for more comprehensive evaluations in the future.

To accomplish this, we have incorporated current production projections and assessed the prospective technological landscape for China's 2030 primary chemicals. We first collected data from both academic literature and gray sources. Then we consulted with industry experts (particularly from Sinopec, and China National Petroleum & Chemical Planning Institute), who provided invaluable commentary and validation of our data and results.

Based on this, we devised a streamlined calculation for 2030 projections. This framework enabled us to project primary chemical production, parametrize the anticipated market assimilation of new technologies (taking into account shifts in feedstock mix, integration of green hydrogen, etc.), and deduce the overarching emission intensity. This factored in various components such as green electricity adoption, shifts in the energy mix, energy efficiency enhancements, and more. Our results further emphasize the compelling need for the integration of green hydrogen, particularly in light of China's chemical industry's current reservations about coal reliance and its dependence on oil and natural gas imports, combined with the young chemical plant assets. We noted the details in the

Data_S2_Flows.xlsx.

Now the paragraph in the Discussion Section reads:

Enhancing energy efficiency stands out as a readily implementable strategy for immediate emission reductions, yet more effective emission reductions necessitate the adoption of alternative green technologies. Using a computational experiment, we projected the 2030 patterns (target year for China's overall carbon peak) and primary chemical production's carbon emissions. These projections were grounded in expected production yields, intensity trends, and technological readiness (See Supplementary Data S2 and Supplementary Notes for details). It suggests that except for ammonia, most primary chemical production would be two to three times as large by 2030 compared to 2017 levels. Without mitigation strategies, emissions in 2030 might surge by 87% (+520 Mt) compared to 2017 levels (Supplementary Figure S12). The upgrade of the electricity grid (energy mix shift), however, could play a significant role in emissions mitigation. The expected reduction of the carbon intensity of supplied electricity[58] from 4.88 to 4.22 tCO₂/tce reduces emissions by a total of 80 Mt compared to what they would be with an unchanged electricity mix. Furthermore, the integration of green hydrogen within coal-to-chemical operations emerges as a promising approach for substantial emission reductions. Our analysis indicates that even a modest integration of green hydrogen—ranging from 0.5 to 5% as projected by experts and institutions for technology readiness by 2030—along with measures including CCUS in the production processes of olefins, methanol, and synthetic ammonia, could mitigate an additional 162 Mt of emissions in 2030, which is nearly a quarter of the levels recorded in 2017 (Supplementary Figure S12).

This highlights the transformative role of green hydrogen, capable of reducing up to 68% of CO₂ emissions from the traditional syngas process[59]. The concentration of China's coal chemical production in regions like Ningxia (Ningdong), Inner Mongolia (Ordos), and Shaanxi (Yulin) synergizes with the substantial renewable energy capabilities, hinting at a promising integration of coal chemical, hydrogen energy, and CCUS in this area (Supplementary Figures S13-15 and Supplementary Notes) [52]. Significantly, China's young production-related assets represent over half of the global chemical industry's total capacity[38]. Such a strategic integration seems more pragmatic than allowing these young chemical infrastructures to become obsolete very soon.

In addition, considering the technology readiness levels, we explain that it is challenging to achieve large-scale implementation of CCUS and biomass-derived feedstock by 2030. While a full-scale quantitative analysis has not been integrated into the primary 2030 scenario for these components, supplemental data detailing their emission reduction potential, as inferred from existing literature, has been incorporated in SI *Data_S2_Flows.xlsx*.

Now the paragraph in the Discussion Section reads:

While some forthcoming technologies show promise, their full-scale implementation might be challenging by 2030. High CO₂ concentrations in chemical production make carbon capture a cost-effective option, yet by 2030, CCUS might offset only 30-50 Mt of CO₂ in the chemical domain in China[60]. PetroChina[36] predicts a 50% cost reduction in China's CCUS by 2060. Considering methanol production through CO₂ hydrogenation could reshape the low-carbon chemical landscape[56], however, supplying abundant, cost-effective green electricity for hydrogen and CCUS generation remains uncertain[12, 18], and larger emission reductions can be attained if this electricity is used to replace coal power plants. Transitioning from fossil hydrocarbons can gain momentum through circular chemical systems[61] and biomass feedstock[14]. With China's recycled plastic pellet output in 2019 standing at 16 Mt, and considering a current recycling efficiency of 30%, this could theoretically displace 12 Mt of coal and crude oil each as primary feedstock[25]. Chemical recycling has the potential to recover valuable materials from waste, but technology complexities suggest that only 11-17% of the aggregate demand might be satisfied by recycled plastics[62]. While biomass-derived chemicals hold promise for emissions reduction, their potential contribution is expected to be only 1-3% by 2030 in China[63] and less than 10% globally by 2050[14], limited by the biomass availability and the rate of investment (See Supplementary Data S2 for the estimated mitigating potentials). The high costs present a formidable barrier. To achieve substantial emissions reductions, we argue that robust policy interventions are critical. These should aim to enhance the economic viability of emerging green technologies and foster their widespread adoption.

At the same time, we have noted the assumption we used in the Supplementary Notes for each process. In acknowledging our study's scope, we recognize the limitations, which in turn outline potential directions for subsequent research. Accordingly, we have revised the concluding paragraph of the discussion:

For future works, broadening the temporal scope and encompassing a wider array of accounts could enhance the understanding of China's chemical sector history. Our projections extend to 2030, offering a basic outline. Yet, for a more layered future outlook, scenarios should consider technology, costs, and practicality, based on the dynamics of materials, energy, and carbon. Future research should also incorporate market dynamics into the process of shaping supply and demand, acknowledging that economic incentives, often guided by policy measures (particularly related to carbon reduction), are important drivers of change in the chemical industry. It's also essential to underscore the importance of demand-side strategies and circular economy principles, especially concerning plastic consumption and disposal. The potential simulation of recycling within the chemical industry is essential but marked by ambiguity [62] and studies are dependent on details such as exact plastics composition, contamination, and technological progress in recycling, which are difficult to capture at the macro level. Given the multifaceted nature of chemical production, focused research on plant-level emissions reduction is also essential. These insights present the potential for a more refined integration of material cycles into integrated

assessment models (IAMs)[64], thereby enriching climate change and fossil resource scenario discussions.

We hope that these modifications and additions serve to address your concerns and enrich the overall contribution of our manuscript.

Following are some detailed remarks:

(4) L108 and following: No need to summarize the results in the introduction.

Response:

Thank you. We have removed these sentences.

(5) P7: Figure 3 and Figure 4 do not contribute to the understanding and can be left out, since detailed data is provided in the supplementary.

Response:

Thank you for the comment. Our intention with Figures 3 and 4 is to offer a more intuitive representation, emphasizing the insights this study provides, specifically the contribution of various chemical routes to distinct products. These figures illustrate how different products depend on varying proportions of fossil feedstock, a detail best highlighted through MFA research.

While we recognize the detailed data provided in the supplementary section, we believe Figures 3 and 4 serve a distinct purpose, especially in spotlighting certain key chemical products. Thus, we would like to retain these figures in the main body of the article.

(6) L313: Missing graphs on cumulated emissions with fractional distributions to products, processes (e.g. cumulated emissions from steam cracking, coal gasification) and feedstock (naphtha/natural gas/coal).

Response:

Thank you for your feedback. We've incorporated three new figures in the Supplementary information (SI) that present a detailed analysis of cumulative carbon emissions based on products, processes, and feedstocks. They are also attached below:

Figure. Cumulative carbon emissions based on products (Figure S9)

Figure. Cumulative carbon emissions based on processes (Figure S10)

Figure. Cumulative carbon emissions based on feedstocks (Figure S11)

(7) Question on the uncertainty analysis methodology: Generally, process product yields and CO2 emissions are associated (e.g. for coal-to-methanol production. Was this considered?

Response:

Thank you for your suggestions. We use the mass balance, i.e., the carbon flow into a process is the carbon in the product plus the emissions. The process-based (direct) emission and the product yields are linked and calculated based on material flow (stoichiometry) analysis. Indirect emissions related to energy, are determined by multiplying external intensity factors by the production yields. In this way, the factor is considered.

We further consulted national standards to determine uncertainty intervals. These standards distinctly categorize emission parameters into "Threshold value", "Standard value", and "Advanced value", each corresponding to facilities of different scales and process efficiencies. Leveraging this framework, we defined the uncertainty intervals. When these intervals are aligned with the output data, it enables us to determine the uncertainty range for emissions at each production stage.

(8) The following publications could be helpful:

<https://doi.org/10.3390/en13184859>

<https://doi.org/10.1016/j.resconrec.2021.106106>

<https://doi.org/10.1016/j.dib.2022.107848>

<https://doi.org/10.1016/j.egypro.2017.03.1821>

Response:

Thank you for recommending these publications. We have consulted the provided literature and incorporated their insights into our quantitative research. We appreciate your constructive feedback and valuable suggestions.

Response to Reviewer #2:

The novelty, hypothesis of the methods, and the main assumptions of the models still need further clarification. Certain issues require addressing before the paper can be accepted for publication.

Response:

Thank you for your remarks about the quality of the study and your suggestions for improving the manuscript. We have carefully revised the manuscript and supplementary documents following your comments below.

Regarding the methods and assumptions, we have made intensive revisions to the Methods Section and Supplementary Notes, which now detail the assumptions necessary for calculating each process.

(1) Title. This work is mainly focused on tracing the non-energy use of fossil fuels and carbon emissions, including plastics, fertilizers, and chemicals as the author mentioned in the main context, not including energy. So it seems that the title is not consistent with what is covered in the main body text.

Response:

Thank you for pointing out the potential inconsistency between the original title and the main content of our work.

To better reflect the study's emphasis on the feedstock use of fossil fuels, particularly in the production of plastics, fertilizers, and chemicals, we have updated the title to: "*Tracing Fossil-Based Plastics, Chemicals and Fertilizers Production in China*".

(2) Abstract. Line 27-30, "Results show China utilizes 5% of the consumed coal, 15% of crude oil, and 7% of natural gas as chemical feedstocks" please indicate the base year and the base value (in energy basis or mass basis) of the quantitative results. Also please ensure consistency throughout the context, e.g., page 3 lines 109-111 and line 145-149.

Response:

Thank you for this. We have revised the sentences to specify both the base year. The energy and mass basis is distinguished by the unit indicated. In the Results section, since we would like to represent the uncertainty, we opted not to round to whole integers. We have ensured the base year and value are consistent throughout the manuscript.

Now the sentence in the abstract reads:

Results show in 2017, the chemical industry used 0.18 Gt of coal, 88.8 Mt of crude oil, and 12.9 Mt of natural gas as feedstock, constituting 5%, 15%, and 7% of China's respective total use.

We have also added a sentence at the end of the Introduction Section to make it clear:

All results presented in this study are based on data from the year 2017.

(3) Line 29, “Coal heavy processes including methanol, ammonia, and PVC plastics production contribute to ~3% of China’s national carbon emissions”---this information is not found in the Results and the Discussion sections, there should be somewhere to explain and discuss this point. Otherwise, it should be presented in the Abstract. Also, since this work is based on gate-to-gate LCA, rather than cradle-to-grave LCA, the rationale of the percentage statement is worth debating.

Response:

Thank you for your suggestion. For consistency with the abstract, we have refined this statement in the abstract to read:

Coal-fed production, including methanol, ammonia, and PVCs contributes to 0.27 Gt carbon emissions (~3% of China’s emissions).

It is also discussed in the CO₂ Emissions section.

Regarding our approach to carbon emissions calculation, as detailed in the original manuscript and *Response #20*, our study utilizes the carbon accounting method. This method concentrates on specific primary chemical production processes, similar to scope 2 emission. While it mirrors the "gate-to-gate" concept in LCA, our research avoids percentage expressions due to potential ambiguity as you suggested, changing to absolute values instead.

(4) Introduction. 4.Line 101, “Chemicals, plastic, and fertilizer production contribute to 10-14% of China’s total emissions, equivalent to” Please provide the base year of these values.

Response:

Thank you for pointing this out and we have indicated the base year as “around 2017”. Accordingly, the revised sentence in the manuscript is:

Estimates show China’s industry accounted for 10-14% of China’s total CO₂ emissions around 2017, equivalent to approximately 4% of global emissions[19, 38-40].

(5) The novelty needs further explanation. Is this the first work to trace fossil-based chemical production? if so, is it worth mentioning, or should explain the novelty in comparison with previous works?

Response:

Thank you for highlighting the need for clarity on the novelty of our work. The revised paragraph now reads:

*In this study, we present a tailored process-based material flow model for China's chemical industry, highlighting the interwoven mass and carbon flows within the expansive (gate-to-gate) production network. **While global chemical flows have been mapped by Levi and Cullen[4], our research uniquely focuses on China's chemical industry, which stands out as the world's largest and is heavily reliant on coal. We further assess the process emissions arising directly from chemical production, alongside energy emissions stemming from the provision of heat, electricity, and steam by using the coefficient-based carbon accounting method (scope 2 accounting). This allows us to pinpoint the production of primary chemical building blocks and identify the most carbon-intensive processes.***

Collecting foundational data for mass and carbon flow operations within the hydrocarbon-based chemical industry is challenging[4, 41]. This study seeks to overcome these data limitations by sourcing and cross-verifying information from various resources, including the yearbook (China Chemical Industry Yearbook[42], Energy Statistics Yearbook[43]), disclosed industrial reports, other databases, and engagements with industry practitioners, all synthesized with an underpinning of chemical engineering knowledge. We also use Monte Carlo simulation[44] to explore potential deviations. All results presented in this study are based on data from the year 2017.

Our analysis also underscores the challenges of decarbonizing China's chemical industry and anticipated shifts in the usage of hydrocarbon resources as primary materials. In essence, our analysis establishes a solid groundwork for future research to examine the broader implications of major structural changes on the energy, chemicals, plastics, and emissions nexus.

(6) Results. Fuel use and feedstock use. Line 125-130, Figure 1, mentioned “In 2017, China consumed 4.7 Gt of fossil hydrocarbons, including 3.9 Gt/yr of coal, 0.6 Gt of crude oil, and 0.2 Gt of natural gas”, and “In 2017, 93% of the coal demand was met by domestic production while 70% of crude oil and 37% of natural gas were imported to meet the domestic demand.” are these figures calculated or quoted statistics? If they are quoted figures, please provide data sources.

Response:

Thank you for your questions. The figures related to fossil fuel consumption, namely "In 2017, China consumed 4.7 Gt of fossil hydrocarbons, including 3.9 Gt/yr of coal, 0.6 Gt of crude oil, and 0.2 Gt of natural gas" are sourced from the China Energy Statistical Yearbook. We have ensured this citation is appropriately referenced in the manuscript.

Regarding the dependence on imports, we calculated them using the same yearbook. For instance, the dependence on crude oil imports was determined based on the net crude oil imports in relation to domestic consumption:

Net crude oil imports = (Import volume: 41,946 - Export volume: 486)

External dependence = Net crude oil imports / Consumption volume: 58,902 =

70.3%.

(Note: The units for the above calculations are all in 10 kt.)

It's important to highlight that our result slightly differs from real-time news data for 2018, which recorded a dependence of 67.5%. This discrepancy may arise from subsequent data validation in statistics. Nonetheless, this slight variation does not impact our main conclusions.

Now the revised paragraph reads:

China's fossil fuel consumption patterns highlighted its reliance on coal and significant import dependency for oil and gas. In 2017, China consumed 4.7 Gt (billion tons, equivalent to petagrams, Pg) of fossil hydrocarbons, including 3.9 Gt·yr⁻¹ of coal, 0.6 Gt·yr⁻¹ of crude oil, and 0.2 Gt·yr⁻¹ of natural gas, as shown in Figure 1. China's natural abundance of coal contrasted with its insufficient oil and gas reserves, leading to the utilization landscape of coal and substantial import needs for crude oil and natural gas. Specifically, a substantial 93% of coal requirements were sourced domestically, aligning with China's rich coal deposits. Conversely, to satisfy its needs, China imported 70% of its crude oil and 37% of its natural gas consumption in 2017 (which varies slightly from 2018 real-time news data, showing 68%^[45] oil and 39%^[46] gas dependence, likely due to subsequent validation).

(7) Line 142, “using material flow modeling, we are able to distinguish between the two constituents of fossil hydrocarbon consumption: fuels for energy production and feedstocks for chemical production”---the sentence is confusing: are the overall fossil consumption, energy production, and feedstocks for chemical production quantified in this work?

Response:

Thank you for your suggestions. We have revised the sentence to make it clear. Now it reads:

Our analysis distinguishes the roles of fossil hydrocarbons in energy versus feedstock for chemical production among the overall fossil fuel consumption in China. According to our material flow modeling, Figure 2 illustrates the distribution of fossil hydrocarbon use in 2017. The majority of fossil hydrocarbons were used for energy, with coal at 95.4(±1)%, oil at 85.1(±0.2)%, and natural gas at 92.9(±1)% based on mass. In contrast, the chemical industry utilized 4.6(±1)% of the consumed coal, 14.9(±0.2)% of the consumed crude oil, and 7.1(±1)% of the consumed natural gas as feedstocks.

(8) Data in Figure 2 is not consistent with the figure in lines 184-186.

Response:

Thank you for your feedback. We have now ensured that the content in both the figures and text-align is consistent. Your reminder prompted us to meticulously review all the data presented in the article for coherence.

(9) Line 144, “Figure 2 depicts a comprehensive utilization structure of fossil hydrocarbons in China. The results show that, Consequently, the chemical industry utilize...as feedstocks”. ---the sentence is confusing, do these results have anything to do with Figure 2? I suppose they should be illustrated in Figure 1. However, even so, it is still unclear how these numbers came out. And on what basis? Energy or mass basis?

Response:

Thank you for your suggestions. We have removed the sentence mentioning Figure 2 here. To clarify, in Figure 1(a), we present data in the energy unit (Exajoule, EJ), and in Figures 1(b) and (c), we use the mass unit (Gigaton/petagram), which is indicated in the charts and legend.

(10) Mapping Chemical Production Flows. I would suggest unifying the units in Figures 1 and 2, all on an energy basis or on a mass basis to avoid confusion.

Response:

Thank you for pointing that out.

In Figure 1(a), we utilize the energy unit (Exajoule, EJ), whereas Figures 1(b), (c), and Figure 2 adopt the mass unit (Gigaton/petagram). The reason for this differentiation in Figure 1 is to provide a comprehensive view of fossil resources' dual attributes – both as energy and material.

Given that the focus of this article is material flow, subsequent sections predominantly use the mass unit. We have specified the units in both the text and accompanying figures for clarity.

(11) lines 184-185 and Line 191, the figures are not consistent with those in Figure 2.

Response:

Thank you for your feedback. We have ensured consistency between the figures and the text and have thoroughly reviewed the article's data for accuracy.

(12) Please improve the quality of Figure 2-4.

Response:

Thank you for the comment. A possible clarity issue might arise from embedding the images directly into the Word document, leading to compression. We will ensure to include the high-resolution versions in our final submission for improved clarity and resolution.

(13) CO₂ Emission. Line 273. In the first paragraph of this section, maybe it is necessary to add one or two sentences to explain the LCA methods (cradle-to-grave or gate-to-gate system boundary) and the selected functional units.

Response:

Thank you for your suggestions.

We've refined and expanded upon the methodology section. Our approach primarily involves process-based carbon accounting, utilizing a coefficient method, instead of the standard cradle-to-gate LCA practices. For clarity, we've set our system boundaries in alignment with the "gate-to-gate" concept from LCA. This has been articulated in the revised section as the "coefficient-based carbon accounting method".

Now in the Introduction Section, the related sentences read:

We further assess the process emissions arising directly from chemical production, alongside energy emissions stemming from the provision of heat, electricity, and steam by using the coefficient-based carbon accounting method (scope 2 accounting). This allows us to pinpoint the production of primary chemical building blocks and identify the most carbon-intensive processes.

In the CO2 Emissions Section, the related sentences read:

We analyzed (gate-to-gate) CO2 emissions in chemical processes, distinguishing between direct process emissions [51] and energy-related emissions from producing the heat, electricity, and steam used in the processes.

(14) Repeat information from lines 283-285 and lines 320-322. If not, please clarify more clearly.

Response:

Thank you for your suggestions. We have rewritten the paragraphs to make it clearer. Now this section reads:

We analyzed (gate-to-gate) CO2 emissions in chemical processes, distinguishing between direct process emissions[51] and energy-related emissions from producing the heat, electricity, and steam used in the processes. Given primary chemicals' pivotal role in the industry, accounting for two-thirds of its total energy demand[1], we focused on their CO2 output. Using material flows and process-based coefficient methods, we break down the stages of chemical processes and estimate the stage-wise carbon intensity of various chemical products, which help identify key emitting processes to target mitigation (Figure 5 and Supplementary Figure S9-11 for breakdowns). We determined that in 2017, process emissions from primary chemicals amounted to 0.30 Gt·yr⁻¹ (±18%) of CO2, while energy-related emissions were slightly higher at 0.34 Gt·yr⁻¹ (±17%).

In examining process emissions, it becomes evident that syngas generation from coal and natural gas, along with the coal-based coking operation for producing calcium carbide, stand out as primary process emission contributors (as detailed in Figure 3). Ammonia production, largely for hydrogen sourcing, is responsible for about half of these emissions, with coal pathways emitting 0.13 Gt·yr⁻¹ (±12%) and steam methane reforming (SMR) accounting for 22.3 Mt·yr⁻¹ (±12%). A third of the emissions are linked to methanol-related pathways, such as methanol-to-

olefins (MTO) and methanol-to-propylene (MTP), with an output of 0.10 Gt-yr⁻¹ ($\pm 21\%$). The conversion of coke to calcium carbide adds another significant amount, approximately 36.5 Mt-yr⁻¹ ($\pm 14\%$). It's significant to note that coal-fed chemical production was responsible for an overwhelming 90% of these emissions, which translates to 2.7(± 0.5)% of China's national CO₂ emissions in 2017.

Coal's inherent low hydrogen-to-carbon (H/C) ratio and energy-per-carbon content result in higher emissions from chemical processes[52] than those based on oil or natural gas. To align with the H/C ratio necessary for methanol production or to obtain hydrogen for ammonia synthesis, coal-derived syngas must be processed through a water-gas shift reaction (WGSR). Specifically, for methanol synthesis, the syngas H/C ratio should be greater than 4 ($H_2/CO > 2$). With coal's H/C ratio at approximately 0.7, additional hydrogen is generated via WGSR by converting carbon monoxide to CO₂. Consequently, only about 40(± 2)% of the carbon in coal, when used solely as feedstock, contributes to methanol production. This limitation, rooted in coal's chemical composition, underscores the challenge of emission mitigation, necessitating alternatives to WGSR for hydrogen generation to reduce emissions. Moreover, although petroleum-based processes are linked to fewer direct emissions, the energy consumed in processes such as cracking and reforming still results in considerable carbon emissions.

Energy-related emissions are predominantly generated from the energy supply necessary for crude oil's cracking and reforming pathways. As shown in Figure 5, steam cracking of ethylene is particularly emissions-intensive, with approximately 1.1 tons ($\pm 18\%$) of CO₂ emitted for every ton of ethylene produced and contribute 16 Mt-yr⁻¹ CO₂ emission. Additionally, the energy-intensive process of BTX (benzene, toluene, xylene) reforming contributes a further 1.9 tons ($\pm 18\%$) of CO₂ for each ton manufactured, totaling 42 Mt-yr⁻¹ CO₂. CO₂ emissions from coal and natural gas chemical processes are also substantial, due to the high-temperature requirements of gasification and the Water-Gas Shift Reaction (WGSR). MTO and MTP processes together contribute to 40 Mt-yr⁻¹. The predominance of coal in China's energy portfolio, accounting for 55% in 2017, exacerbates these emissions, as evidenced by a comparison with other global regions in Supplementary Data S2. A prompt shift towards a more sustainable energy mix and the possible electrification of chemical processes could markedly diminish these energy-related emissions[14, 18]. The extent of emissions reduction is determined by the rapidity of the system-wide energy transition and the optimization of energy efficiency specifically within the chemical sector.

(15) In the abstract, the author mentioned “Coal-heavy processes contribute to 3% of China’s national carbon emissions”, this information should be explained and discussed in the results section.

Response:

Thank you for highlighting this. We have revised the relevant sections to better reflect the information mentioned in the abstract. The revised paragraph reads:

In examining process emissions, it becomes evident that syngas generation from coal and natural gas, along with the coal-based coking operation for producing calcium carbide, stand out as primary process emission contributors (as detailed in Figure 3). Ammonia production, largely for hydrogen sourcing, is responsible for about half of these emissions, with coal pathways emitting 0.13 Gt·yr⁻¹ ($\pm 12\%$) and steam methane reforming (SMR) accounting for 22.3 Mt·yr⁻¹ ($\pm 12\%$). A third of the emissions are linked to methanol-related pathways, such as methanol-to-olefins (MTO) and methanol-to-propylene (MTP), with an output of 0.10 Gt·yr⁻¹ ($\pm 21\%$). The conversion of coke to calcium carbide adds another significant amount, approximately 36.5 Mt·yr⁻¹ ($\pm 14\%$). **It's significant to note that coal-fed chemical production was responsible for an overwhelming 90% of these emissions, which translates to 2.7(± 0.5)% of China's national CO₂ emissions in 2017.**

For greater clarity, we have also refined the statement in the abstract:

Coal-fed production, including methanol, ammonia, and PVCs contributes to 0.27 Gt carbon emissions (~3% of China's emissions).

(16) The carbon intensity of different chemical products are quantified as the author mentioned, then it might be helpful to illustrate the stage-wised carbon intensity of different chemical product to identify the key emission processes.

Response:

Thank you for highlighting this message. The detailed results are added in *Supplemental Data_S2_Flows.xlsx*. We have further supplemented three figures calculating cumulative carbon emission breakdown by product, process, and feedstock.

Figure S1. Cumulative carbon emissions breakdown by product

Figure S2 Cumulative carbon emissions breakdown by process

Figure S3. Cumulative carbon emissions breakdown by feedstock

We have revised the paragraph:

*We analyzed (gate-to-gate) CO₂ emissions in chemical processes, distinguishing between direct process emissions[51] and energy-related emissions from producing the heat, electricity, and steam used in the processes. Given primary chemicals' pivotal role in the industry, accounting for two-thirds of its total energy demand[1], we focused on their CO₂ output. **Using material flows and process-based coefficient methods, we break down the stages of chemical processes and estimate the stage-wise carbon intensity of various chemical products, which help identify key emitting processes to target mitigation (Figure 5 and Supplementary Figure S9-11 for breakdowns).** We determined that in 2017, process emissions from primary chemicals amounted to 0.30 Gt·yr⁻¹ (±18%) of CO₂, while energy-related emissions were slightly higher at 0.34 Gt·yr⁻¹ (±17%).*

(17) I would suggest to compare the carbon intensity of China's chemical products and the European's or other places.

Response:

Thank you for your suggestion. We have included this comparison in Supplementary Information *Data_S2_Flows.xlsx* (Sheet 6 Emission intensity comparison), citing databases such as Ecoinvent, PlasticsEurope, and IPCC.

However, it is worth noting that most LCA databases mainly focus on cradle-to-gate emission intensity, whereas our study is based on a gate-to-gate approach. In general, China tends to have higher emission factors than Europe, mainly due to its coal-based energy mix (used to generate electricity, heat, and steam). Now it is also noted in the main text:

The predominance of coal in China's energy portfolio, accounting for 55% in 2017, exacerbates these emissions, as evidenced by a comparison with other global regions in Supplementary Data S2.

(18) Discussion. Limitations should be discussed.

Response:

Thank you for your suggestions to further discuss the limitations. Following your suggestion, we have revised the Methods Section and expanded the Discussion Section to address their points more comprehensively.

- Assumptions made for calculation for each process are noted in Supplementary Notes.
- Key limitations, which concurrently point to future research directions. They are included in the revised Discussion Section, which now reads:

For future works, broadening the temporal scope and encompassing a wider array of accounts could enhance the understanding of China's chemical sector history. Our projections extend to 2030, offering a basic outline. Yet, for a more layered future outlook, scenarios should consider technology, costs, and practicality, based on the dynamics of materials, energy, and carbon. Future research should also incorporate market dynamics into the process of shaping supply and demand, acknowledging that economic incentives, often guided by policy measures (particularly related to carbon reduction), are important drivers of change in the chemical industry. It's also essential to underscore the importance of demand-side strategies and circular economy principles, especially concerning plastic consumption and disposal. The potential simulation of recycling within the chemical industry is essential but marked by ambiguity [62] and studies are dependent on details such as exact plastics composition, contamination, and technological progress in recycling, which are difficult to capture at the macro level. Given the multifaceted nature of chemical production, focused research on plant-level emissions reduction is also essential. These insights present the potential for a more refined integration of material cycles into integrated assessment models (IAMs)[64], thereby enriching climate change and fossil resource scenario discussions.

(19) Methods. The method section needs further explanation: In your mass-balance model, how did you trace the use of different types of fossil fuels for different

chemical products? Did you get the chemical product yield (in the year 2017) first, and then reverse the raw material (coal, oil, and gas) consumption based on mass balances? If so, then how did you assume and rationalize the assumption of the proportion of different raw material production routes? I think it would be more clear if the author could explain the overall idea of the methods and the key assumptions first, and then go into detail to explain the mass balances of different routes.

Response:

Thank you for your suggestions.

Thank you for highlighting this. Your understanding of our methodology is accurate. In the chemical industry, fossil raw materials are primarily converted into primary chemicals, which subsequently undergo downstream processing to produce various products. By analyzing the output of these primary chemicals and employing coefficients associated with the production process, we were able to reverse-engineer the consumption of raw materials.

The distribution across different raw material production routes isn't arbitrary; it's grounded in actual data mainly derived from the "China Chemical Industry Yearbook" and further validation with experts. You may find these specifics in the "Proportion" column of sheets 1.1 and 2.2 in *Supporting Information Data S2*.

To enhance clarity, we have incorporated detailed explanations within our tables that illuminate the context and basis of various data points. In response to your feedback on elucidating our calculation approach, we've made revisions in the "Method" section, specifically under the subheading "*Calculate Feedstock Inputs and Carbon Emissions*", and please kindly find the updated parts in the manuscript. We have also conducted intensive revisions to the Supplementary Notes, which now detail the assumptions necessary for calculating each process.

(20) In your LCA-based carbon emission calculation models. Please clarify the system boundary, functional units, and allocation basis (energy-based, mass-based, or economic-based, and why so?) should be explained. Key assumptions should be listed or explained.

Response:

Thank you for pushing us further to clarify this point. In this study, we focused on the production of primary chemicals. These specific processes are not only highly energy-intensive but also account for approximately two-thirds of the chemical industry's total energy consumption.

For our calculations, we adopted a carbon accounting approach based on the coefficient method. To illustrate, let's examine the production process of synthetic ammonia:

Figure. Diagram Illustrating Carbon Emission Accounting in the Ammonia Synthesis Process

System Boundary: The above figure outlines the system boundary chosen, which is the production process of coal-to-synthetic ammonia. Inputs encompass materials such as coal, oxygen, nitrogen, and water, along with energy forms like electricity and heat. The outputs consist of the primary product, synthetic ammonia, and various non-CO₂ substances. The emissions derived from this process include energy-related emissions and process-related emissions, each of which will be further defined and calculated below.

Carbon Emissions and Their Calculation:

- **Energy Emissions:** These represent the carbon emissions embedded in the electricity and steam utilized during the process. They are derived by multiplying the consumed electricity and heat by a consistent electricity/heat emission factor obtained from factory data.
- **Process Emissions:** These are by-products of the chemical reactions and are determined by the reaction's input-output relationships and stoichiometry.
- **Total Emissions:** This is the aggregate of the energy and process emissions. The carbon intensity of a product is defined as the total emissions divided by the main product's quantity (measured in tCO₂ per ton of product).
- For processes producing multiple products (like steam cracking), we compute the overall emissions and then distribute them to the respective products based on their mass. Consequently, for such multi-output processes, the carbon emission intensity (tCO₂/t product) remains consistent across products.

To summarize, for the calculation of carbon emissions from each primary chemical production process, this study utilizes the carbon emission accounting (coefficient) method, equivalent to Scope 2 emission. Our results specifically address carbon dioxide emissions, rather than general greenhouse gas emissions.

For clarity and ease of reader understanding, we incorporate the "gate-to-gate" concept from LCA; however, our method does not align with the standard steps applied in the typical LCA practices. To prevent any misinterpretation, we refer to our approach as the "coefficient-based carbon accounting method" throughout this paper.

We've incorporated the aforementioned description of carbon emission accounting for primary chemicals into both the Method Section and the Supplementary Notes (SI) of the paper.

(21) Supplemental material. On page 15, the calculation is confusing. It seems the functional unit of this calculation is tCO₂/t Naphtha (the input raw material), rather than tCO₂/t product.

Response:

Thank you for pushing us to be clearer.

In the steam cracking process, based on mass conservation, the total mass of the input raw materials equals the combined mass of all outputs, which include ethylene, propylene, C4 components, C6-C8 components, and others. Here, "others" doesn't mean mass losses but refers to components like hydrogen and methane. In our calculations, we allocate carbon emissions based on mass. The carbon emissions assigned to the products will match those allocated to the raw materials. Here, for our method, we specifically allocate carbon emissions to the products.

We have updated the CO₂ accounting section in the Supplementary Notes to provide more details about the calculation process. We have also added explanations and formulas to make our methodology clearer:

$$\begin{aligned} E_{e,cracking} &= \frac{\text{total electricity consumption}}{\text{product quantity}} \\ &= \frac{0.051 \text{ tce}}{(1 + 0.5 + 0.3 + 0.5 + 0.7) \text{ t product}} = 0.017 \text{ tce/t product} \\ E_{h,cracking} &= \frac{\text{total heat consumption}}{\text{product quantity}} = \frac{1.068 \text{ tce}}{(1 + 0.5 + 0.3 + 0.5 + 0.7) \text{ t product}} \\ &= 0.356 \text{ tce/t product} \end{aligned}$$

(22) Supplementary note Page 14, the electricity emission factor seems too high for China's coal-fired fleets.

Response:

Thank you for pointing this out.

We have double-checked our underlying source. Here we used the average emission factor of the power grid as published by the Chinese government (Climate Change of the People's Republic of China Second Biennial Update Report (2018):

<https://www.mee.gov.cn/ywgz/ydqhbh/wsqtzk/201907/P020190701765971866571.pdf>; and Notice from Ministry of Ecology and Environment of People's Republic of China on the key tasks related to the management of corporate greenhouse gas emission reports in 2022:

https://www.mee.gov.cn/xxgk2018/xxgk/xxgk06/202203/t20220315_971468.htm). For reference, this factor was 0.6101 tCO₂/MWh in 2015 and reduced to 0.5839 tCO₂/MWh by 2020. In this way, we can derive the compound annual growth rate (CAGR) of the emission factor as:

$$CAGR = \sqrt[2020-2015]{\frac{0.5839}{0.6101}} - 1 = -0.874\%$$

So the 2017 carbon emission factor can be calculated as:

$$0.6101 \times (1 - 0.874\%)^2 = 0.5995 \text{ tCO}_2/\text{MWh}$$

In this study, due to the non-uniform original data units and confidentiality of some data, we opted to use tons of standard coal (tce) as the consistent unit for our emission coefficients. The "ton of standard coal" is a unit frequently employed in Chinese industrial production, symbolizing any energy source with a calorific value of 7×10^6 kcal. Thus, the conversions are:

$$1 \text{ tce} = 7 \times 10^6 \text{ kcal} = 29307.6 \text{ MJ} = 8141 \text{ kWh}$$

Transforming this emission factor into the unit of tCO₂/tce, we obtain the emission coefficient grounded on tce as:

$$Q_e = 0.5995 \text{ tCO}_2/\text{MWh} \times 8.141 \text{ MWh/tce} = 4.880 \text{ t CO}_2/\text{tce}$$

We also apply this CAGR method to project the emission factor of the grid in 2030.

Response to Reviewer #3:

(1) The manuscript submitted by Jiang et al. investigates the use of fossil fuels for the production of chemicals in China. The analysis includes mapping the fossil fuels feedstock in the chemical industry and the computation of the associated scope 1 (process) and scope 2 (energy input) CO₂ emissions. Authors investigate both the production of primary chemicals (e.g. methanol) and of final chemical products (e.g. polyethylene). Generally speaking, reconstructing the carbon cycle of the chemical industry, especially when done for a diverse and large country as China, is a complex task that requires a clear and smart methodology, and a large number of input data from scientific, industrial, and institutional literature. Per se, the topic would fit well the journal aim and scope, and would be highly relevant for a large scientific community and for several policy makers. Generally speaking, the manuscript is clearly written (English is ok) and to the point, but it misses a compelling underlying narrative. It is for example unclear if authors aim at providing a picture of the carbon flow in the chemical industry, if they want to discuss possible solutions in a world of constrained fossil fuels, or both. Figures and plots are of good quality and help the understanding of the paper. However, in the reviewer's opinion, the paper does not reach the quality for publication in Nature Communication. There are several aspects that support this conclusion.

Response:

Thank you for your comprehensive feedback.

We fully agree with your saying that *"reconstructing the carbon cycle of the chemical industry, especially for a vast country like China"*. It necessitates a precise methodology and extensive data from localized sources. We appreciate your recognition of this challenge and significance, emphasizing the meticulous approach our study necessitated.

The main goal of our study is to present a detailed overview of material and carbon flows in China's chemical industry, with a focus on modeling these flows. Our material flow model is central to the research presented in this article. It overcomes the data deficiencies and provides a clearer understanding of the complex interactions within chemical processes at a macro level. Based on this model, we seek to discuss potential strategies for sustainability. The close link between material and carbon flows, particularly in processes that convert fossil hydrocarbons, highlights the importance of extending carbon accounting on material flow. This method offers a comprehensive perspective and identifies critical processes for informing targeted policy-making.

Taking into account your feedback and suggestions from other reviewers, we have intensively made improvements to our manuscript and supplemented our data. Also, we have enriched our content by incorporating an experimental projection up to 2030. This not only underscores the adaptability of our model but also has implications for the potential emission reductions that specific strategies may achieve in the short to medium term. Those are based on engaging with petrochemical industry experts and revisiting academic and industry literature.

Your comments and suggestions have been instrumental in refining our paper. We have acted on each point raised to enhance the overall quality and coherence. We hope that these modifications and additions serve to address your concerns. Please kindly review them below.

(2) First, the analysis of the results is very shallow; it is unclear what the main findings and take-home messages are. The paper does not provide any conclusion, thus leaving the reader with a feeling of unfinished work.

Response:

Thank you for your comments prompting us to clarify our take-home messages and conclusions. We have elaborated on key data points in the Results Section and intensively revised the Discussion Section. Please kindly review the updated manuscript. Here is a concise summary:

Main findings and take-home messages:

- While the role of chemical feedstocks in the consumption of fossil fuel is often undervalued, their increasing demand is a critical factor in the growth of global hydrocarbon use, which deserves more attention.
- 5% to 15% of China's hydrocarbon supply is used as chemical feedstock.
- As a result of China's insufficient domestic production of oil and gas, the country uses coal as a feedstock for many chemicals, causing high CO₂ emissions, especially for olefins, ammonia, and PVC production.
- Only up to 40% of coal's carbon is converted into chemicals, while the rest is oxidized and emitted as CO₂.
- Our projections for 2030 indicate a potential two to threefold growth in primary chemical production compared to 2017. Without mitigation strategies, emissions might surge by 2030.
- The integration of green hydrogen and renewable electricity is crucial for achieving meaningful carbon reduction in China's chemical sector.
- Coal's importance in China's energy and food security suggests that integrating renewables (power-to-X) with existing coal infrastructures is more pragmatic than rendering these infrastructures obsolete.
- To achieve substantial emissions reductions, we argue that robust policy interventions are critical. These should aim to enhance the economic viability of emerging green technologies and foster their widespread adoption.

We hope this updated Results and Discussion Section and clarified response addresses your concerns and thank you for your comments that improved the quality of our work.

(3) In the text, authors mostly report numbers presented in the figures without a (or a very limited) critical analysis of them. They also try to identify solutions for the reduction of scope 1 and 2 CO₂ emissions (biomass, CCUS,..), but they do that in an un-organized, qualitative way spread across different paragraphs of the results and

discussion. For example, it is rather disappointing to read at line 382 of page 11 that authors identify CCUS as a negative emissions technology, which is indeed wrong unless applied to biomass-derived CO₂ and associated to permanent carbon storage. For publication in this journal, authors should provide a more insightful and detailed analysis.

Response:

Thank you for your constructive suggestions. To address your primary concerns, we have undertaken the following key revisions:

- Improved writing and enhanced the analysis in the discussion and results sections.
- Provided a focused experimental analysis concerning potential emission reductions to 2030, offering broader implications.
- Corrected the expressions of CCUS and included technology readiness levels in the discussion.

Regarding the implications of carbon emissions and potential reduction solutions, while the central aim of our study is to illustrate a comprehensive material flow and key carbon emission landscape in chemical production, we acknowledge that a quantitative analysis explores and quantifies potential emission reduction trajectories within the Chinese chemical industry, offering broader implications. We have added an experimental analysis to provide a preliminary quantification of potential emission reductions in the short term (the target year 2030 for China's carbon peak). Our revised discussion now states:

Enhancing energy efficiency stands out as a readily implementable strategy for immediate emission reductions, yet more effective emission reductions necessitate the adoption of alternative green technologies. Using a computational experiment, we projected the 2030 patterns (target year for China's overall carbon peak) and primary chemical production's carbon emissions. These projections were grounded in expected production yields, intensity trends, and technological readiness (See Supplementary Data S2 and Supplementary Notes for details). It suggests that except for ammonia, most primary chemical production would be two to three times as large by 2030 compared to 2017 levels. Without mitigation strategies, emissions in 2030 might surge by 87% (+520 Mt) compared to 2017 levels (Supplementary Figure S12). The upgrade of the electricity grid (energy mix shift), however, could play a significant role in emissions mitigation. The expected reduction of the carbon intensity of supplied electricity[58] from 4.88 to 4.22 tCO₂/tce reduces emissions by a total of 80 Mt compared to what they would be with an unchanged electricity mix. Furthermore, the integration of green hydrogen within coal-to-chemical operations emerges as a promising approach for substantial emission reductions. Our analysis indicates that even a modest integration of green hydrogen—ranging from 0.5 to 5% as projected by experts and institutions for technology readiness by 2030—along with measures including CCUS in the production processes of olefins, methanol, and synthetic ammonia, could mitigate an additional 162 Mt of emissions in 2030, which is nearly a quarter of the levels recorded in 2017 (Supplementary Figure S12).

This highlights the transformative role of green hydrogen, capable of reducing up to 68% of CO₂ emissions from the traditional syngas process[59]. The concentration of China's coal chemical production in regions like Ningxia (Ningdong), Inner Mongolia (Ordos), and Shaanxi (Yulin) synergizes with the substantial renewable energy capabilities, hinting at a promising integration of coal chemical, hydrogen energy, and CCUS in this area (Supplementary Figures S13-15 and Supplementary Notes) [52]. Significantly, China's young production-related assets represent over half of the global chemical industry's total capacity[38]. Such a strategic integration seems more pragmatic than allowing these young chemical infrastructures to become obsolete very soon.

We appreciate you pointing out the CCUS issue. We've rectified this and further included the technology readiness levels in our discussion. Our revised discussion section on this topic reads:

While some forthcoming technologies show promise, their full-scale implementation might be challenging by 2030. High CO₂ concentrations in chemical production make carbon capture a cost-effective option, yet by 2030, CCUS might offset only 30-50 Mt of CO₂ in the chemical domain in China[60]. PetroChina[36] predicts a 50% cost reduction in China's CCUS by 2060. Considering methanol production through CO₂ hydrogenation could reshape the low-carbon chemical landscape[56], however, supplying abundant, cost-effective green electricity for hydrogen and CCUS generation remains uncertain[12, 18], and larger emission reductions can be attained if this electricity is used to replace coal power plants. Transitioning from fossil hydrocarbons can gain momentum through circular chemical systems[61] and biomass feedstock[14]. With China's recycled plastic pellet output in 2019 standing at 16 Mt, and considering a current recycling efficiency of 30%, this could theoretically displace 12 Mt of coal and crude oil each as primary feedstock[25]. Chemical recycling has the potential to recover valuable materials from waste, but technology complexities suggest that only 11-17% of the aggregate demand might be satisfied by recycled plastics[62]. While biomass-derived chemicals hold promise for emissions reduction, their potential contribution is expected to be only 1-3% by 2030 in China[63] and less than 10% globally by 2050[14], limited by the biomass availability and the rate of investment (See Supplementary Data S2 for the estimated mitigating potentials). The high costs present a formidable barrier. To achieve substantial emissions reductions, we argue that robust policy interventions are critical. These should aim to enhance the economic viability of emerging green technologies and foster their widespread adoption.

Furthermore, acknowledging our study's scope, we have further discussed the future works that could be based on our study:

For future works, broadening the temporal scope and encompassing a wider array of accounts could enhance the understanding of China's chemical sector history. Our projections extend to 2030, offering a basic outline. Yet, for a more layered

future outlook, scenarios should consider technology, costs, and practicality, based on the dynamics of materials, energy, and carbon. Future research should also incorporate market dynamics into the process of shaping supply and demand, acknowledging that economic incentives, often guided by policy measures (particularly related to carbon reduction), are important drivers of change in the chemical industry. It's also essential to underscore the importance of demand-side strategies and circular economy principles, especially concerning plastic consumption and disposal. The potential simulation of recycling within the chemical industry is essential but marked by ambiguity [62] and studies are dependent on details such as exact plastics composition, contamination, and technological progress in recycling, which are difficult to capture at the macro level. Given the multifaceted nature of chemical production, focused research on plant-level emissions reduction is also essential. These insights present the potential for a more refined integration of material cycles into integrated assessment models (IAMs)[64], thereby enriching climate change and fossil resource scenario discussions.

We hope that these modifications and additions serve to address your concerns and enrich the overall contribution of our manuscript.

(4) Second, the method and associated data input are very unclear. While authors provide a substantial supplementary information file and data files, it remains almost impossible to judge if the numbers presented in the paper are likely to be correct or not. While I do recognize that similar analyses, i.e. closing the carbon cycle of a national chemical industry, is very difficult and almost impossible to prove, I believe authors should have done a much better job in illustrating and explaining the method, which remains shallow and unclear in both the manuscript and the SI.

Response:

Thank you for your suggestions. we have revisited and restructured both the method and Supplementary Information (SI) sections of our manuscript.

To further clarify our approach, we have incorporated schematic diagrams detailing system boundaries and the overall workflow. These additions aim to provide a more lucid illustration of our methodology and offer a step-by-step demonstration of our calculations through specific examples.

Figure. System boundary and workflow of this study (Figure 7 in main texts).

New modification to the SI:

- We have detailed the various chemical products and demonstrated their relationships within the upstream-downstream network.
- We have enhanced the guidelines for computing fossil fuel inputs and carbon dioxide emissions.
- To aid readers' comprehension, calculation examples have been included.
- Additional information regarding the process, as well as the interrelation among the SI, Method, and Data_S2, has been incorporated where necessary.
- Assumptions needed for calculation are noted for each process.

New modification to Data_S2:

- We have provided clarifications for table titles and notes where essential, ensuring readers can clearly interpret the data's context.
- Data regarding the primary chemical production process and its connection to downstream Node Chemicals has been included.
- Updates have been made to the carbon emission comparison data, and new data on carbon emission scenarios has been integrated.
- We have refreshed the reference sources indicating whether the data is publicly available and, if so, how to access it.

(5) Moreover, authors use several (grey) sources for their analysis, for example in the second excel data file, that are difficult to trace online or unspecified. For instance, I tried to look for ref.1 (China Petroleum and Chemical Industry Federation, China Chemical Industry Yearbook 2017-2018. China Chemical Information Center: Beijing, 2019) but could only find the CPCIF annual report (2019) which did not contain the data shown in the excel file. Ref. 5 in the same excel file is not a real reference: it namely says 'Customs statistics online search platform'. Authors should make the original data sources available to the reader, not only the data itself. As for the method, I find the explanation of the method rather poor and superficial. Authors could start their revision with a block diagram containing the different steps of the method.

Response:

Thank you for pointing out these concerns. We understand the challenges that international scholars may face when trying to access specific data sources, particularly those in other languages or that require distinct database subscriptions. Our research involved significant efforts to gather the necessary data, and our primary objective was to bridge these data gaps.

For greater transparency and to facilitate data verification, we have now enhanced the Supplementary Data_S2_Flows.xlsx by including titles in Chinese, their English translations, and direct access links. Furthermore, we have documented each data flow within this file.

We have also revalidated every data link. We mark each link according to its accessibility status (e.g., open access, subscription required, or private).

Recognizing that researchers may not access specific data due to language or subscription barriers, we hope to provide convenience. Should any researcher face issues accessing particular data, we're more than willing to assist upon request.

To clarify the two sources that you mentioned:

The "Chemical Industry Yearbook" is available through the China National Knowledge Infrastructure (CNKI) online link (<http://60.16.24.131/CSYDMirror/Trade/yearbook/single/N2021010132?z=Z012>), a platform Tsinghua University subscribes to. For clarity, we have attached screenshots of the relevant data below, and updated information in Data_S2_Flows.xlsx.

Page from the "Chemical Industry Yearbook" in the CNKI Database (original one; see translation below):

[redacted]

Page from the "Chemical Industry Yearbook" in the CNKI Database. (translation):

[redacted]

The data is archived in the "Chemical Industry Yearbook" within the CNKI Database (see attached screenshots):

表 9 2017 年中国石油和化学工业及相关行业主要产品产量

产品名称	2016	2017	2017/2016 增长 (%)
原油	19942.4	19150.6	-4.0
天然气 (亿 m ³)	1358.6	1474.2	8.5
煤层气 (亿 m ³)	64.9	70.2	8.2
液化天然气	724.6	829.0	14.4
原油加工量	54078.6	56777.3	5.0
成品油	34776.0	35825.1	3.0
汽油	12895.6	13276.2	3.0
煤油	3983.8	4230.9	6.2
柴油	17896.6	18318.0	2.4
燃料油	2711.7	2693.4	-0.7
石脑油	3318.8	3401.2	2.5
液化石油气	3518.9	3677.3	4.5
石油焦	2628.5	2721.7	3.5
石油沥青	3283.4	3980.0	21.2
焦炭	44599.9	43142.6	-3.3
原煤	333804.9	344545.6	3.2
煤气 (亿 m ³)	10225.2	10626.9	3.9
发电量 (亿 kWh)	59387.7	62758.2	5.7
火力发电量 (亿 kWh)	44103.3	46114.6	4.6
水力发电量 (亿 kWh)	10460.5	10818.8	3.4
核能发电量 (亿 kWh)	2132.9	2480.7	16.3
风力发电量 (亿 kWh)	2220.7	2695.4	21.4
太阳能发电量 (亿 kWh)	469.3	647.5	38.0
硫铁矿石 (折含 S 35%)	1412.3	1458.8	3.3
磷矿石 (折含 P ₂ O ₅ 30%)	12241.6	12313.2	0.6
合成氨 (无水氨)	5190.8	4785.8	-7.8
化肥总计 (折纯)	6227.5	6065.2	-2.6
氮肥 (折含 N 100%)	4012.9	3834.9	-4.4
尿素 (折含 N 100%)	2897.2	2629.4	-9.2
磷肥 (折含 P ₂ O ₅ 100%)	1616.8	1627.4	0.7
钾肥 (折含 K ₂ O 100%)	597.8	599.7	0.3
磷酸一铵 (实物量)	1939.7	1895.8	-2.3
磷酸二铵 (实物量)	1561.3	1502.4	-3.8
化学农药原药 (折 100%)	322.1	294.1	-8.7
杀虫剂原药 (折 100%)	54.0	59.7	10.5
杀菌剂原药 (折 100%)	14.9	17.0	14.6
除草剂原药 (折 100%)	142.6	114.8	-19.5
硫酸 (折 100%)	8547.0	8694.2	1.7
盐酸 (含 HCl 31%)	786.8	777.8	-1.2
浓硝酸 (折 100%)	268.5	278.1	3.6
氢氧化钠 (烧碱, 折 100%)	3193.8	3365.2	5.4
离子膜法烧碱 (折 100%)	2759.0	2868.4	4.0
碳酸钠 (纯碱)	2550.8	2677.1	5.0
碳化钙 (电石, 折 300L/kg)	2406.0	2447.3	1.7
单晶硅 (t)	106000.7	165056.5	55.7
多晶硅 (t)	263522.1	298032.1	13.1

The "Customs statistics online search platform" (<http://stats.customs.gov.cn/indexEn>) requires users to enter specific criteria such as the HS code and timeframe to obtain the desired data. We have attached screenshots of the platform and the resulting data for reference. The necessary HS codes are also detailed in Data_S2_Flows.xlsx.

Customs statistics online search platform (using Polyethylene HS39011000 as an example):

[redacted]

Customs statistics online search platform (Screenshot):

[redacted]

Also following your suggestion, we have enhanced our Supplementary Information (SI) and datasets, and introduced block diagrams (Figure 7 in Methods) to clearly outline our methodology and diagrams that outline the material transformation networks of ethylene, propylene, C4, BTX, Ammonia, Methanol and Calcium Carbide as well as an overview (Supplementary Figure S16-21). Please kindly review the Methods, SI, and Supplementary Datasets.

Figure. System boundary and workflow of this study (Figure 7 in main texts).

Figure Upstream Material Transformation (Figure S16)

Figure Downstream Material Transformation: Ethylene (Figure S17)

Figure Downstream Material Transformation: Propylene (Figure S18)

Figure Downstream Material Transformation: C4 and BTX (Figure S19)

Figure Downstream Material Transformation: Ammonia (Figure S20)

Figure Downstream Material Transformation: Methanol and Calcium Carbide (Figure S21)

(6) Another disappointing point concerns the fact that in the SI authors refer to an Experimental procedures section, which however I could not find in any of the files sent with the submission. For example, authors mention that they have done a Monte Carlo simulation to tackle uncertainties, but there is no explanation of it (they again cites an Experimental procedures section which cannot be found).

Response:

Thank you for the comments and the attention to detail. We apologize for the oversight concerning the referenced "Experimental Procedures section". This was indeed a typo. The intended reference is the "Method Section". We have made the necessary corrections.

To ensure clarity and completeness, we have expanded upon and provided more comprehensive details about the uncertainty analysis, which can be found in the revised manuscript's "Methods Section" and "Supplementary information". The updated section reads as follows:

In the Method Section

Uncertainty analysis

Refineries and chemical plants even in the same area often have different process parameters. Therefore, an uncertainty analysis was used to determine the potential deviation of results. Here we used Monte Carlo simulations to evaluate the uncertainties in our material flow model [43,67]. In our MFA modeling process, there are mainly two types of data inputs: output of the products (yield data) and process-based coefficient data (product/feedstock ratio, etc.). We assumed the yield data (from the official yearbook [41,42]) are relatively accurate and we focused on the uncertainties of process coefficients. All uncertain coefficients and parameters in the Monte Carlo simulation are described by normally distributed independent random variables due to the limits of process data samples. We assumed the baseline value is the initial guess – setting it as the mean of normal distribution. The uncertainty range of each individual data point was given based on the quality of the sources and the suggested uncertainty range obtained from factory data, literature, and experts' experiences (see Supplementary Notes and Supplementary Data S2 for details). Further, we used $[1/6 \cdot \text{deviation range}]$ (for example, $0.4/6$ for a deviation range of $\pm 20\%$) as the standard deviation in this approach to cover 99.7% of the range of change (within 3 standard deviations in a normal distribution). Finally, we combined different uncertainties to arrive at the final confidence value by running 20,000 Monte Carlo Simulations. We chose the 95% certainty level where the lower-bound and upper-bound are noted as 2.5% and 97.5% percentiles [68]. We do not indicate the uncertainty range for data sourced from Statistical Yearbooks and results with a deviation interval of less than 0.5%.

In the Supplementary information:

Uncertainty analysis

Diverse refineries and chemical plants, even within the same vicinity, often operate under differing process parameters. We employed an uncertainty analysis to gauge the potential deviation in results, using Monte Carlo simulations for uncertainties in our material flow model [32,33].

In our MFA modeling process, there are mainly two types of data inputs:

- Product Outputs (Yield Data): Assumed to be accurate as sourced from the official yearbook.*
- Process-Based Coefficient Data: Including product-to-feedstock ratios, which were our primary focus due to their uncertainties.*

We employed normally distributed independent random variables to describe all uncertain coefficients and parameters, basing the baseline value on our initial guess. This value was set as the mean of the normal distribution. The uncertainty range was ascertained from quality sources, factory data, literature, and expert opinions (see Supplementary Notes and Supplementary Datasets S2 for details). We used 20,000 Monte Carlo simulations to deduce a final confidence value, choosing the 95% certainty level for results. In a word, we used $[1/6 \cdot \text{deviation range}]$ (for example, $0.4/6$ for a deviation range of $\pm 20\%$) as the standard

deviation in this approach to cover 99.7% of the range of change (within 3 standard deviations in a normal distribution). Specific data from Statistical Yearbooks and results with a deviation of less than 0.5% were excluded from the uncertainty range.

For primary chemicals production—primarily the conversion of hydrocarbons from fossil fuels—uncertainty was higher. National standards provided values for these processes, categorizing them into "threshold", "standard", and "advanced" energy/fossil resource consumption levels. Each level denotes a different efficiency tier in production:

- *Threshold value: Minimum efficiency for new production ventures. The level that new production capacity, such as new construction, renovation, expansion projects, etc., must meet. Its value should, in theory, represent the last 20% of the industry's energy efficiency level.*
- *Standard value: The required efficiency level for existing businesses. The value should be based on the elimination of a certain percentage of existing high-energy-consuming backward production capacity. Approximately 20% of backward products and production capacity should be eliminated as a result of energy-saving transformation.*
- *Advanced value: It benchmarks the leading energy efficiency level of the same type of production.*

Given the mandatory nature of these national standards, we consider them highly representative of reality. Chemical reactions in downstream production, being well-defined, present lower uncertainty.

The uncertainty range data are given in four ways:

- *For primary chemicals production and processes directly consuming fossil hydrocarbons, we use data from the National Standard of the People's Republic of China (see Table S3-S12 below). The series of standards Norm of energy consumption per unit product gives the Threshold value, Standard value, and advanced value of energy consumption per unit product. We use the larger range of Threshold value and advanced value from Standard value. If there is no standard for a process, use 5% for the uncertainty range.*
- *For PE, PP, PVC, PS, and butadiene rubber production, the material consumption is close to the thermodynamic limit. Therefore, we consider the uncertainty range as 1%.*
- *For fertilizer and coal-based vinyl chloride production, different formulations can bring about large changes in raw material inputs. Therefore, we consider the uncertainty range as 10%.*
- *For other processes, a general uncertainty range of 5% is assumed.*

For a detailed uncertainty range of each point, refer to Data_S2_Flows.xlsx.

(7) To conclude, I find the paper and the presented analysis of interest for the journal and worth considering for publication; however, the scientific quality of the

submitted manuscript and the associated supplementary information and data files do not reach the minimum standard required for a scientific journal.

Response:

Thank you for providing feedback on our manuscript and noting the topic and content of the study.

Following all your suggestions, we have made significant improvements, including refining the narrative, revising the discussion sections, and adding more detailed information in the supplemental documents.

We consulted with domain experts to validate our findings and methodologies, further strengthening the scientific robustness of our work. We believe that the revised version aligns more closely with your expectations. We deeply appreciate your constructive suggestions.

Reviewers' Comments:

Reviewer #1:

Remarks to the Author:

The presented manuscript provides relevant insights into the production structure and emission sources of the unique and globally most relevant chemical industry in China, enabling the international science community (and other stakeholders) to gain a better understanding of a field that is usually hard to access, which makes it a valuable contribution for publication.

The discussion of the results (from line 344) is barely insightful and not much more than a summary, but I also wouldn't really know what specifically to address further, as I see the value mostly in the supply of data and the balance for further investigations.

The perspective discussion until 2030 is tentative and holds little value in the current state. The quantified assessment should be either further clarified and better explained in the main text, or it could be left out or shortened and reduced to a final outlook.

The methodology as presented is hard to follow, despite that it appears that all required information are provided mostly in the SI. Please restructure the chapter with the objective in mind that it should be possible to understand the general approach and structure without looking into the SI. What is especially not clear to me: how where conflicts in the applied reference data (used feedstock, produced chemical quantities etc.) harmonized, i.e. what data was fixed and what was adjusted, how was decided what was deemed true for the model?

Consequently, despite the potential and scientific value of the results and provided insights, major improvements on the manuscript structure and accessibility would be necessary to provide the quality for publication.

Reviewer #4:

Remarks to the Author:

This submission investigated the non-energy use of fossil fuels and CO₂ emissions in China. Results show in 2017, the chemical industry used 0.18 Gt of coal, 88.8 Mt of crude oil, and 12.9 Mt of natural gas as feedstock, constituting 5%, 15%, and 7% of China's respective total use. Coal-fed production of methanol, ammonia, and PVCs contributes to 0.27 Gt CO₂ emissions (~3% of China's emissions). As China seeks to balance high CO₂ emissions of coal-fed production with import dependence on oil and gas, improving energy efficiency and coupling green hydrogen emerges as attractive alternatives for decarbonization.

The response to the comments and suggestions of previous reviewers have been addressed. All Figures are excellent and will serve as very useful references for researchers. Similarly, the Supplementary Notes and Data Tables are clear and well-organized. The authors present rich underlying data. They mentioned that they have engaged with many industry professionals. It is very good to fill gaps in existing literature and the well-known data gap in this field and increase the robustness of their results. The paper also touches upon the future transformation of China's chemical industry and its projected carbon emissions by 2030, which is important for subsequent research.

In addition, the manuscript could further benefit from a more detailed analysis in the discussion section, particularly highlighting the specific challenges that lie ahead for the Chinese chemical industry. Also, the authors touch upon differing perspectives on industry transformation between chemical practitioners and environmental academics. This aspect could be emphasized more in the discussion, offering a richer and deeper understanding of the sector's future trajectory.

The following articles can be followed to catch more information about the emissions of coal production and coal chemical industry.

Remarkable spatial disparity of life cycle inventory for coal production in China. *Environmental Science & Technology*, 2023, 57(41): 15443-5453. <https://doi.org/10.1021/acs.est.3c01860>

Intensive carbon dioxide emissions of coal chemical industry in China. *Applied Energy*, 2019, 236: 540-550. <https://doi.org/10.1016/j.apenergy.2018.12.022>

Response to Reviewer #1:

(1) The presented manuscript provides relevant insights into the production structure and emission sources of the unique and globally most relevant chemical industry in China, enabling the international science community (and other stakeholders) to gain a better understanding of a field that is usually hard to access, which makes it a valuable contribution for publication.

Response:

Thank you for your insightful feedback and recognition of our study's contribution to providing a comprehensive understanding of the complex production structure and emission sources within China's chemical industry—a sector that has been challenging to analyze due to its complexity and global significance.

Following your constructive suggestions, we have carefully revised our manuscript to enhance clarity and enrich the discussion and methodological sections. We hope the detailed information offered could provide more insights and references to the international scientific community across various fields and those with an interest in this topic. Please kindly find the detailed revision and response below.

(2) The discussion of the results (from line 344) is barely insightful and not much more than a summary, but I also wouldn't really know what specifically to address further, as I see the value mostly in the supply of data and the balance for further investigations.

Response:

Thank you for your constructive feedback, which has encouraged us to deepen the discussion in our manuscript. We are grateful for your acknowledgment of the value our article brings with its mass-balance model, addressing the sophisticated material flows and carbon accounting within China's chemical industry.

Following your suggestions, we have rewritten the discussion section to particularly highlight the critical role of mass balance modeling in capturing system details and targeted emission reduction efforts within the chemical industry. We have also enriched this section with our reflections on the model's development and analytical process. Now this part of the discussion reads:

While the role of chemical feedstocks in the consumption of fossil fuel is often underappreciated, their increasing demand is a critical factor in the growth of global hydrocarbon use[1,14,55]. The connection between production and consumption in energy/climate scenario models has historically been obscured by insufficient attention to non-energy uses and the complexities of fossil fuel production networks, complicating the assessment of demand-side measure effectiveness. Despite the thorough exploration of supply-side measures, evidenced by many scenario analyses, they alone prove inadequate⁴. Implementing circular economy principles, demand reduction strategies, and integrated energy scenario assessments necessitates a comprehensive detail of material cycles in chemical production[56]. Levi and

Cullen[4] emphasized the essential need for precise mass balances and clear documentation of the industry's primary material flows, vital for accurately predicting the impact of mitigation strategies further along the value chain.

Reflecting on the establishment of a macro-balance within the petrochemical industry reveals several insights. Initially, mapping the network of chemical substances within a region's industrial framework is both essential and complex, going beyond merely cataloging key products and processes. It also requires an understanding of their intricate interrelations, diverging from traditional Material Flow Analysis (MFA) due to the sophisticated nature of the chemical metabolism. Insights from this work, along with previous efforts[1,4,14,57,58] could provide references, yet obtaining robust production and process data remains challenging. It often entails corroborating information from diverse sources, including statistical reports, literature, and industry studies, with regional technological diversity adding complexity to data collection, especially for smaller-scale downstream productions. Inevitably, a mass balance on this scale will be affected by inconsistencies among divergent data sources. Our methodology, incorporating a loss term to address yield discrepancies, aims to preserve the integrity of existing data while acknowledging its limitations. Such an extensive mass balance necessitates ongoing collaboration between academics and industry practitioners to improve data availability and transparency.

Moreover, we have emphasized the significance of engaging stakeholders, particularly industry practitioners, in our discussions. Our findings reveal divergent perspectives on future scenarios between chemical engineers and environmental scholars, underscoring the necessity of integrating diverse viewpoints in our analysis. Specifically, if the industry is not challenged, they will not find ways to achieve decarbonization. And this part now reads:

Engagement with industry experts was essential for validating our model's assumptions and results. The dialogue with practitioners on decarbonizing China's chemical sector revealed a difference in outlook compared to the long-term sustainability focus of environmental experts. This difference, emphasizing immediate capabilities and economic considerations, highlights the need to challenge industry norms and incentives to encourage proactive decarbonization efforts. For instance, the Rocky Mountain Institute's projections[39] for the adoption of green hydrogen or CCS in coal-based capacities by 2030 (30%) were seen as overly optimistic by industry experts, who anticipated only a 5% integration. Moreover, continuous monitoring is needed, given the anticipated significant capacity expansions in China and the United States, with Asia and the Middle East driving long-term growth[1]. Regional differences in process pathways, such as the reliance on ethane in North America and the Middle East versus the use of naphtha for high-value chemicals in Asia and Europe, require focused attention in modeling. These processes are major contributors to carbon emissions and play a significant

role in downstream economics, emphasizing the need for region-specific analyses. Additionally, the petrochemical industry's dynamic nature, considering the maturity of chemical processes, necessitates regular updates to the model, with data collection, for example, every five years to ensure its continued relevance and accuracy.

Following another of your suggestions below (#3) we have streamlined the outlook for 2030, making it more concise and focused, and thereby also illustrating the model's adaptability. This includes a discussion on the strategic integration of technologies like green hydrogen, which plays a vital role in both scenario analysis and actual implementation. Now this part reads:

Such a mass-balance framework can extend to a simple scenario outlook, as we explore China's 2030 based on industry expert estimates (see Supplementary Data S2, Notes, and Figure S12). Our findings suggest that, except ammonia, the output of most primary chemicals could double or triple from 2017 levels by 2030. Without strategic interventions, emissions might surge by 87% (+520 Mt) from 2017. Transitioning to a cleaner energy mix could significantly lower emissions. An expected reduction in the carbon intensity of electricity from 4.88 to 4.22 tCO₂/tce could decrease emissions by 80 Mt compared to maintaining the current energy mix. Furthermore, integrating green hydrogen into coal-to-chemical processes emerges as a key strategy for substantial emission reductions. A modest adoption rate of 0.5 to 5% by 2030, as forecasted by various experts, along with the application of carbon capture, utilization, and storage (CCUS) in producing olefins, methanol, and synthetic ammonia, could cut emissions by an additional 162 Mt in 2030, nearly a quarter of the 2017 levels.

This underscores green hydrogen's potential to reduce CO₂ emissions by up to 68% from the conventional syngas process[59]. The geographic concentration of China's coal chemical production in areas like Ningxia, Inner Mongolia, and Shaanxi, combined with a significant renewable energy potential, suggests a promising integration of coal chemical processes, hydrogen energy, and CCUS[54] (refer to Supplementary Figures S14-15 and Notes). Notably, China's relatively young manufacturing assets represent over half of the global chemical industry's capacity[39]. Such a strategic integration seems more pragmatic than allowing these young chemical infrastructures to become obsolete quickly. However, emerging technologies offer potential, and their widescale adoption by 2030 poses challenges (detailed further in Supplemental Notes 2.5-2.6).

We conclude by addressing the policy implications stemming from our findings and outlining potential future research directions. Now this part reads:

Chinese policymakers[60] are considering measures such as enhancing energy efficiency, advancing electrification, integrating green hydrogen and CCUS, transitioning to lighter hydrocarbons, 'crude oil-to-chemical' integration, and

promoting the transformation towards a circular economy, etc. Our material flow model provides detailed information, from broad process linkages to specific policy and technology intervention points[19]. However, realizing these technological advancements depends on updating existing and constructing new chemical facilities (such as green hydrogen production). Additionally, ensuring an adequate supply of low-carbon electricity is crucial for supporting electrification and CCUS efforts[18], necessitating integrated planning across the energy and chemical sectors. On the demand side, fostering changes in consumer behavior and lifestyles is vital for real demand reduction and the advancement of a circular economy, especially in altering plastic consumption patterns for substantial environmental benefits.

For future analyses, the integration of supply-side and demand-side strategies within a physics-based framework enables direct quantification of new technology adoption, considering timelines, infrastructure compatibility, investment needs, and integration ease. It also accounts for the influence of market dynamics, especially the effects of policy-driven incentives and costs on carbon emission reduction efforts. The potential simulation of recycling flows within the chemical industry is necessary but marked by ambiguity[61] that only 11-17% of the demand might be satisfied by recycled plastics. Studies are dependent on details such as exact plastic composition, contamination, and technological progress in recycling, which are difficult to capture at the macro level. Given the multifaceted nature of chemical production, focused research on plant-level emissions reduction is also practical. These insights present the potential for a more refined integration of material cycles into integrated assessment models (IAMs)[56], thereby enriching climate change and fossil hydrocarbon scenario discussions. This holistic approach underscores the need for continuous model monitoring, data-driven scientific support[62], and proactive policymaker and practitioner engagement to enhance carbon emission management, facilitate carbon data accessibility, and guide enterprises towards energy-efficient and low-carbon practices.

We sincerely thank you for your constructive suggestions, which have been instrumental in enhancing the discussion's depth and clarity.

(3) The perspective discussion until 2030 is tentative and holds little value in the current state. The quantified assessment should be either further clarified and better explained in the main text, or it could be left out or shortened and reduced to a final outlook.

Response:

Thank you for your suggestion regarding the 2030 outlook. Inspired by your previous suggestion (#2), we have refined the discussion section to present a more focused outlook towards 2030. This revision underscores the necessity of quantitatively integrating forward-looking technologies such as green hydrogen.

We also have emphasized the adaptability of our model, demonstrating its potential for broader application in future scenario analyses. In the future, we plan to undertake more extensive research in this direction, building on this groundwork. These enhancements and our plans for future exploration are detailed in the last paragraphs dedicated to future work. For detailed changes, please refer to our earlier response regarding the discussion or in the manuscript.

(4) The methodology as presented is hard to follow, despite that it appears that all required information are provided mostly in the SI. Please restructure the chapter with the objective in mind that it should be possible to understand the general approach and structure without looking into the SI. What is especially not clear to me: how where conflicts in the applied reference data (used feedstock, produced chemical quantities etc.) harmonized, i.e. what data was fixed and what was adjusted, how was decided what was deemed true for the model?

Response:

Thank you for highlighting the need for clarity in our methodology. We appreciate the opportunity to make our approach more accessible and understandable, bridging the gap in academia and industry. In light of your feedback, we have undertaken a comprehensive revision of the Methods section and Supplementary Information (SI).

To improve readability and comprehension, we have introduced a new segment at the beginning of the Methods section that summarizes the study's overarching workflow and principles. This addition is complemented by an illustrative diagram in Figure 6, designed to provide a clearer understanding of our modeling methodology. Now it reads:

The study models fossil hydrocarbon processing in three stages: total consumption split to fuel and feedstock (blue), conversion from fossil feedstock to primary chemicals (yellow), and further manufacturing to downstream products (green), as shown in Figure 6. Our primary focus is on the yellow and green sections, emphasizing fossil hydrocarbon feedstock use.

Figure 6. System boundary and workflow of this study.

Following is the overall workflow and principles of this study:

(1) Determine total fossil feedstock inputs (yellow): There are 21 processes for producing 7 primary chemicals (ethylene, propylene, C4 olefins, BTX aromatics, ammonia, methanol, and calcium carbide) and 5 processes directly utilize fossil hydrocarbons as feedstock. We categorize these 26 processes into three distinct groups and determine the total fossil feedstock inputs based on the production yield of each chemical and the process-based coefficients and stoichiometry.

(2) Split energy and feedstock use of hydrocarbons (blue): This is addressed through a mass balance approach, based on the result from the previous step and energy-balancing data from the China Energy Statistics Yearbook[60].

(3) Connect primary chemicals with downstream production (green): There are 48 chemicals and 51 processes for the three tiers of downstream production. In this model, each production process is treated as a node, where mass balance relationships for reactants and products are established based on the production yield of each chemical and the process-based coefficients and stoichiometry. See Figure S16-21 for their chemical interlinks. Acknowledging the inherent yield inaccuracies in macroscopic data modeling, we do not use an enforced balance like harmonization techniques in STAN. To be specific, for inconsistencies, we assess the quality and reliability of the data, including expert consultations, to determine whether input or output data is more dependable. Subsequently, we take the remaining values into imbalance items, which are documented alongside other 'losses' to maintain data integrity while recognizing its inherent limitations.

(4) Trace production mix and feedstock intensity (yellow/green): Based on the interwoven material flow results from the previous steps, we can trace the detailed production mix (i.e., the proportion of three production processes, coal-based, oil-based, or gas-based, of the total domestic production amount) and feedstock intensity (the average amount of fossil feedstock used per ton of product) of each chemical. These values build quantitative bridges between the final products and fossil feedstocks.

(5) Calculate carbon emissions for primary chemical production processes in 2017 and quantify future mitigation potential in 2030 (yellow): We calculate the scope 2 carbon emissions of the 21 primary chemical production processes in China, in 2017. The process-related emission (or direct emission) is the CO₂ as the by-product of the chemical reaction. The energy-related emission (or indirect emission) is the equivalent CO₂ of the electricity and heat consumption in the production process. We further conduct a simplified scenario analysis to quantify the short-term mitigation potential for China's primary chemical production in 2030. Three distinct scenarios include the baseline scenario (only upscaling the chemical demand), grid improvement scenario (enhanced energy structure), and technology scenario (adoption of low-carbon technologies).

(6) Uncertainty analysis: We apply Monte Carlo simulations to evaluate the

uncertainties in our material flow model. We categorize the process-based coefficients into four types and assign different levels of uncertainties. We arrive at the final confidence value by running 20,000 rounds.

Further, we have refined the Methods section to clearly differentiate between the carbon emission calculations and the mass balance modeling, aiming for greater transparency. These refinements are mirrored in the SI for consistency.

Addressing your concerns regarding data harmonization, we have enhanced the explanation of our approach to managing inconsistencies within the mass balance. Rather than applying reinforced harmonization techniques akin to those in STAN – the material flow analysis software, we introduced an 'imbalance item (loss)' to address discrepancies. Specifically, we evaluate the data's quality and reliability, often consulting experts, to discern the more reliable data between inputs and outputs. Then the resulting discrepancies are allocated to 'imbalance items', recorded with other 'losses', to achieve balance and preserve data integrity, while acknowledging limitations. This methodological clarification is now explicitly detailed in the revised sections of both the Methods and Discussion.

(5) Consequently, despite the potential and scientific value of the results and provided insights, major improvements on the manuscript structure and accessibility would be necessary to provide the quality for publication.

Response:

Thank you for acknowledging the scientific value of our research. We appreciate your constructive feedback which greatly improves the quality of our manuscript! We are grateful for your guidance and hope the revised manuscript meets your expectations. Thank you again for your valuable input!

Response to Reviewer #4:

(1) This submission investigated the non-energy use of fossil fuels and CO₂ emissions in China. Results show in 2017, the chemical industry used 0.18 Gt of coal, 88.8 Mt of crude oil, and 12.9 Mt of natural gas as feedstock, constituting 5%, 15%, and 7% of China's respective total use. Coal-fed production of methanol, ammonia, and PVCs contributes to 0.27 Gt CO₂ emissions (~3% of China's emissions). As China seeks to balance high CO₂ emissions of coal-fed production with import dependence on oil and gas, improving energy efficiency and coupling green hydrogen emerges as attractive alternatives for decarbonization.

The response to the comments and suggestions of previous reviewers have been addressed. All Figures are excellent and will serve as very useful references for researchers. Similarly, the Supplementary Notes and Data Tables are clear and well-organized. The authors present rich underlying data. They mentioned that they have engaged with many industry professionals. It is very good to fill gaps in existing literature and the well-known data gap in this field and increase the robustness of their results. The paper also touches upon the future transformation of China's chemical industry and its projected carbon emissions by 2030, which is important for subsequent research.

Response:

Thank you for your remarks about the quality of the study and your suggestions for improving the manuscript. We have carefully revised it following your comments below.

(2) In addition, the manuscript could further benefit from a more detailed analysis in the discussion section, particularly highlighting the specific challenges that lie ahead for the Chinese chemical industry. Also, the authors touch upon differing perspectives on industry transformation between chemical practitioners and environmental academics. This aspect could be emphasized more in the discussion, offering a richer and deeper understanding of the sector's future trajectory.

Response:

Thank you for your constructive suggestions on enhancing the discussion section of our manuscript.

Following your suggestions, first, we refined the presentation of our 2030 outlook, to highlight both the opportunities and challenges facing the transformation and emission reduction efforts within China's chemical industry, with a particular focus on its significant coal dependency. We have underscored the potential role of emerging technologies, such as green hydrogen and CCUS, within this context. However, we acknowledge the substantial challenges associated with their deployment rates and the future availability of green power supply. Now the part reads:

Such a mass-balance framework can extend to a simple scenario outlook, as we explore China's 2030 based on industry expert estimates (see

Supplementary Data S2, Notes, and Figure S12). Our findings suggest that, except ammonia, the output of most primary chemicals could double or triple from 2017 levels by 2030. Without strategic interventions, emissions might surge by 87% (+520 Mt) from 2017. Transitioning to a cleaner energy mix could significantly lower emissions. An expected reduction in the carbon intensity of electricity from 4.88 to 4.22 tCO₂/tce could decrease emissions by 80 Mt compared to maintaining the current energy mix. Furthermore, integrating green hydrogen into coal-to-chemical processes emerges as a key strategy for substantial emission reductions. A modest adoption rate of 0.5 to 5% by 2030, as forecasted by various experts, along with the application of carbon capture, utilization, and storage (CCUS) in producing olefins, methanol, and synthetic ammonia, could cut emissions by an additional 162 Mt in 2030, nearly a quarter of the 2017 levels.

This underscores green hydrogen's potential to reduce CO₂ emissions by up to 68% from the conventional syngas process[59]. The geographic concentration of China's coal chemical production in areas like Ningxia, Inner Mongolia, and Shaanxi, combined with a significant renewable energy potential, suggests a promising integration of coal chemical processes, hydrogen energy, and CCUS⁵⁴ (refer to Supplementary Figures S14-15 and Notes). Notably, China's relatively young manufacturing assets represent over half of the global chemical industry's capacity³⁹. Such a strategic integration seems more pragmatic than allowing these young chemical infrastructures to become obsolete quickly. However, emerging technologies offer potential, and their widescale adoption by 2030 poses challenges (detailed further in Supplemental Notes 2.5-2.6).

Chinese policymakers[60] are considering measures such as enhancing energy efficiency, advancing electrification, integrating green hydrogen and CCUS, transitioning to lighter hydrocarbons, 'crude oil-to-chemical' integration, and promoting the transformation towards a circular economy, etc. Our material flow model provides detailed information, from broad process linkages to specific policy and technology intervention points[19]. However, realizing these technological advancements depends on updating existing and constructing new chemical facilities (such as green hydrogen production). Additionally, ensuring an adequate supply of low-carbon electricity is crucial for supporting electrification and CCUS efforts[18], necessitating integrated planning across the energy and chemical sectors. On the demand side, fostering changes in consumer behavior and lifestyles is vital for real demand reduction and the advancement of a circular economy, especially in altering plastic consumption patterns for substantial environmental benefits.

Then, following your suggestion, we have expanded our discussion on stakeholder engagement. The involvement of industry experts is crucial not only for validating the assumptions and results of our model but also for ensuring a holistic understanding of the sector's dynamics.

Moreover, during our interactions with practitioners, we found that chemical engineers and environmental scholars may hold divergent views on future scenarios, highlighting the importance of incorporating various perspectives into our analysis. Specifically, if the industry is not challenged, they will not find ways to achieve decarbonization. Consequently, we have revised the Discussion section to reflect these insights:

Engagement with industry experts was essential for validating our model's assumptions and results. The dialogue with practitioners on decarbonizing China's chemical sector revealed a difference in outlook compared to the long-term sustainability focus of environmental experts. This difference, emphasizing immediate capabilities and economic considerations, highlights the need to challenge industry norms and incentives to encourage proactive decarbonization efforts. For instance, the Rocky Mountain Institute's projections³⁹ for the adoption of green hydrogen or CCS in coal-based capacities by 2030 (30%) were seen as overly optimistic by industry experts, who anticipated only a 5% integration. Moreover, continuous monitoring is needed, given the anticipated significant capacity expansions in China and the United States, with Asia and the Middle East driving long-term growth[1]. Regional differences in process pathways, such as the reliance on ethane in North America and the Middle East versus the use of naphtha for high-value chemicals in Asia and Europe, require focused attention in modeling. These processes are major contributors to carbon emissions and play a significant role in downstream economics, emphasizing the need for region-specific analyses. Additionally, the petrochemical industry's dynamic nature, considering the maturity of chemical processes, necessitates regular updates to the model, with data collection, for example, every five years to ensure its continued relevance and accuracy.

(3) The following articles can be followed to catch more information about the emissions of coal production and coal chemical industry.

Remarkable spatial disparity of life cycle inventory for coal production in China. Environmental Science & Technology, 2023, 57(41): 15443-5453. <https://doi.org/10.1021/acs.est.3c01860>

Intensive carbon dioxide emissions of coal chemical industry in China. Applied Energy, 2019, 236: 540-550. <https://doi.org/10.1016/j.apenergy.2018.12.022>

Response:

Thank you for recommending these insightful articles on coal production and the emissions associated with the coal chemical industry. We have thoroughly reviewed these studies and compared their findings to our own, ensuring the consistency of our results with the established data.

We have integrated this comparison into our manuscript, particularly in the Supplemental Dataset S2 'Emission Intensity Comparison' section. We value your constructive feedback and the opportunity to improve the quality of our work with these valuable suggestions.

Reviewers' Comments:

Reviewer #1:

Remarks to the Author:

All open questions were addressed, thank you for the significant effort.